# Discffusion: Discriminative Diffusion Models as Few-shot Vision and Language Learners

**Xuehai He**[1]  **Weixi Feng**[2]  **Tsu-Jui Fu**[2]  **Varun Jampani**[3]  **Arjun Akula**[4]  **Pradyumna Narayana**[4]
**Sugato Basu**[4]  **William Yang Wang**[2]  **Xin Eric Wang**[1]
[1]**UC Santa Cruz,** [2]**UC Santa Barbara,** [3]**Stability AI,** [4]**Google**
`{xhe89,xwang366}@ucsc.edu`

Reviewed on OpenReview: `https://openreview.net/forum?id=GtnipgAomT`

## Abstract

Diffusion models, such as Stable Diffusion (Rombach et al., 2022a), have shown incredible performance on text-to-image generation. Since text-to-image generation often requires models to generate visual concepts with fine-grained details and attributes specified in text prompts, can we leverage the powerful representations learned by pre-trained diffusion models for discriminative tasks such as image-text matching? To answer this question, we propose a novel approach, Discriminative Stable Diffusion (Discffusion), which turns pre-trained text-to-image diffusion models into few-shot discriminative learners. Our approach uses the cross-attention score of a Stable Diffusion model to capture the mutual influence between visual and textual information and fine-tune the model via a new attention-based prompt learning to perform image-text matching. By comparing Discffusion with state-of-the-art methods on several benchmark datasets, we demonstrate the potential of using pre-trained diffusion models for discriminative tasks with superior results on few-shot image-text matching.

## 1 Introduction

> *"What I Cannot Create, I Do Not Understand."*
>
> *Richard Feynman*

This quote by Richard Feynman perfectly captures the essence of human learning techniques. In the context of machine learning, it can be interpreted as the ability to generate images given text prompts is a strong indicator of understanding and matching between visual and textual information. Despite the success of various methods in the image-text matching task (Karpathy & Fei-Fei, 2015; Lee et al., 2018), there is still a need for more advanced models that can better capture the fine-grained details, spatial relationships, and compositionality. Meanwhile, diffusion models (Sohl-Dickstein et al., 2015a; Rombach et al., 2022a) have been shown to produce high-quality and diverse images from text descriptions. Therefore, in this paper, we investigate the idea of leveraging the power of pre-trained Diffusion Models, specifically the state-of-the-art text-to-image generative model—Stable Diffusion (Rombach et al., 2022a), for the discriminative image-text matching task, as shown in Figure 1. The success of Stable Diffusion in generative tasks suggests that it has a strong understanding of the relationship between visual and textual information, and we aim to harness this understanding for image-text matching tasks.

The key advantages of using Stable Diffusion for image-text matching are two folds: first, Stable Diffusion uses a VQVAE (Kingma & Welling, 2013; Van DenOord et al., 2017) and cross-attention layers in its

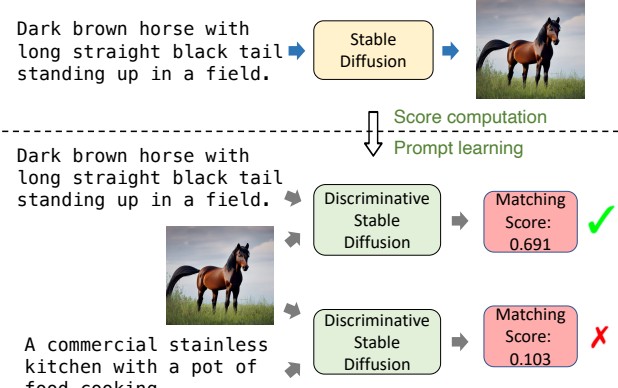

Figure 1: The upper subfigure in the teaser image illustrates the ability of Stable Diffusion to generate realistic images given a text prompt. The bottom subfigure illustrates the process of our proposed method, Discriminative Stable Diffusion (Discffusion), for utilizing Stable Diffusion for the image-text matching task. Discffusion can output a matching score for a given text prompt and image, with a higher score indicating a stronger match.

architecture, which provides strong compressed representations and shed information about the alignment of data from different modalities. Second, Stable Diffusion, through proper adaptations, demonstrates a capacity to comprehend spatial relations (Wu et al., 2023; Derakhshani et al., 2023) and to discern fine-grained, disentangled concepts (Matsunaga et al., 2022; Rambhatla & Misra, 2023). It can generate images that closely align with the specifics of textual prompts, while traditional vision and language model such as CLIP (Radford et al., 2021), which are pre-trained on discriminative tasks, predominantly facilitate coarse-grained, contextual image-text alignment. Such models lack the capability to perform compositional matching at finer granularity, specifically failing to achieve detailed cross-modal alignment at the region-word level (Jiang et al., 2022).

However, to efficiently adapt Stable Diffusion to the image-text matching task, two key challenges need to be addressed: (1) how to disentangle the degree of alignment between the image and text from the latent space of Stable Diffusion? In text-to-image generation, the model is trained to generate an image that is semantically consistent with a given text prompt. However, in image-text matching, the task is to determine the degree of alignment between a given image and text. Therefore, it is important to disentangle the degree of alignment between the image and text in the latent space of Stable Diffusion, to effectively use it for image-text matching; (2) How to efficiently adapt the model in a few-shot setting. Adapting a text-to-image generation model such as Stable Diffusion for image-text matching involves transitioning the model from a generative to a discriminative task. This shift presents significant challenges due to the differences in task requirements and underlying model architectures.

To address these challenges, we propose the Discriminative Stable Diffusion (Discffusion) method, which includes two key ideas: (1) identifying and leveraging attention scores from the selected cross-attention maps as the matching score and (2) employing attention-based prompt learning for model fine-tuning.

Our contributions in this paper are threefold:

- We conduct a pioneering study that repurposes latent text-to-image generation diffusion models—originally designed for generative tasks—for discriminative tasks such as image-text matching.

- We introduce a novel method that leverages cross-attention maps from Stable Diffusion across multiple layers, combined with attention-based prompt learning, to address the image-text matching task.

- We demonstrate the effectiveness of our approach through experimental evaluation on Compositional Visual Genome (Jiang et al., 2022), RefCOCOg (Yu et al., 2016), Winoground (Thrush et al., 2022), and VL-checklist (Zhao et al., 2022) datasets for image-text matching. We further demonstrate the

adaptability of our method by applying it to the visual question answering task, where it excels on the VQAv2 (Antol et al., 2015) dataset.

## 2    Related Work

**Diffusion Probabilistic Models (DPMs)**    Diffusion probabilistic models (DPMs) have been widely used as generative models for images in recent years. These models, which include diffusion (Sohl-Dickstein et al., 2015b) and score-based generative models (Song & Ermon, 2019), have been shown to outperform generative adversarial networks (GANs) (Goodfellow et al., 2014) in many cases (Baranchuk et al., 2021). In the past two years, significant progress has been made in the development of DPMs, with a focus on improving sampling techniques such as classifier-free guidance (Ho & Salimans, 2021). DPMs are typically implemented using convolutional U-Net architectures (Ronneberger et al., 2015a) which contain cross-attention layers. Hertz et al. (2022a) finds that replacing attention maps in the cross-attention module of text-to-image generation diffusion models can edit image attributes. Just scaling the attention maps of the respective word can adjust the effect of a particular word in the prompt. Feng et al. (2022) demonstrates that one can retain the compositional semantics in the generated image by manipulating the cross-attention. Kumari et al. (2022) proposes to fine-tune the key and value mapping from text to latent features in the cross-attention layers of text-to-image diffusion model to compose multiple new concepts in the image. In the context of image-text matching, the attention scores between the text and image representations in the DPMs can reflect the degree of alignment between them.

**Few-shot Learning for Vision and Language Tasks**    Vision and Language discriminative models pre-trained on large-scale image-text pairs have demonstrated great potential in multimodal representation learning (Jia et al., 2021; Yao et al., 2021; Yuan et al., 2021; Radford et al., 2021; Liu et al., 2024). Among them, CLIP (Radford et al., 2021) benefits from 400M curated data and defines various prompt templates to carry out zero-shot image classification. Like CLIP, several different few-shot learners were proposed. GPT (Brown et al., 2020), as a strong few-shot learner, is capable of performing a new language task by learning from only a few training instances. Frozen (Tsimpoukelli et al., 2021) is developed based on GPT and made into a multimodal few-shot learner by expanding the soft prompting to include a collection of images and text. The concept of prompt learning (Schick & Schütze, 2020) has been widely explored in natural language processing (NLP) and computer vision. It allows pre-trained models to adapt to various downstream tasks with minimal number of data by introducing a small prompt layer (Schick & Schütze, 2020; Liu et al., 2021). In the context of image-text matching, prompt learning has been used to fine-tune pre-trained models for the task (He et al., 2022b). In our work, instead of adding learnable prompts over the inputs or between transformer layers (Jia et al., 2022), we introduce learnable prompts over the attention layers. In our paper, our primary research question is the adaptation of pre-trained generative diffusion models into discriminative models for specific tasks. This focus is driven by the challenges and opportunities presented by utilizing diffusion-based processes in a discriminative setting, specifically for the image-text matching task, which has distinct characteristics compared to the modeling approaches mentioned above.

**Generative Models for Discriminative Tasks**    There has been a significant amount of research on using generative models for discriminative tasks in the past decades. Ng & Jordan (2001) compare the discriminative classifier with generative classifier. Ranzato et al. (2011) apply deep generative models to the recognition task. For diffusion models, recently, Li et al. (2023a); Clark & Jaini (2023) propose to use pre-trained diffusion models for zero-shot classification. Wei et al. (2023) formulate diffusion models as masked autoencoders and achieves state-of-the-art classification accuracy on video tasks. Different from these works, we are the first to explore the use of cross-attention maps in pre-trained diffusion models for discriminative tasks, specifically the image-text matching task. Another line of works use diffusion models as data source and then training a discriminative model on the synthetic data generated from it (He et al., 2022a; Jahanian et al., 2021; Zhang et al., 2021). Differs from these works, our approach emphasizes the direct adaptation of generative diffusion models, leveraging their pre-existing structures and knowledge without the need to generate synthetic data.

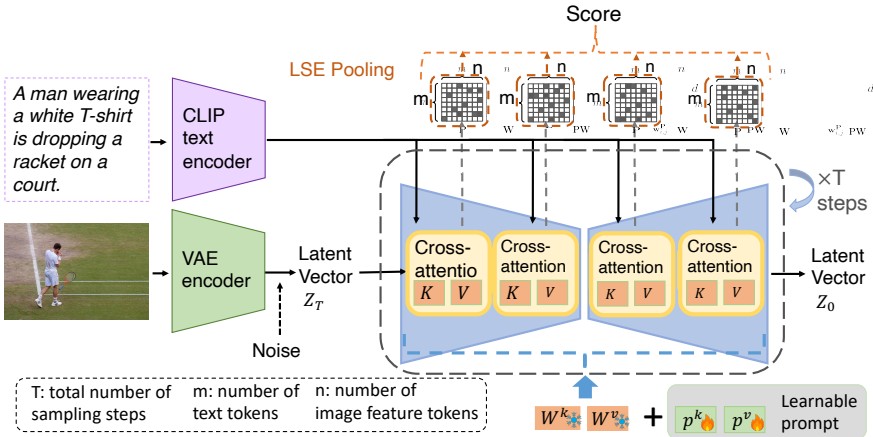

Figure 2: The overview of our Discriminative Stable Diffusion framework, which measures how much the given images and texts matched use the cross-attention mechanism in the Stable Diffusion. Discriminative Stable Diffusion added learnable prompt over attention matrices (red boxes). The learnable prompt will receive gradients during training and updated, while the pretrained weights are fixed. The layer index in $m$ and $n$ is dropped for simplicity.

## 3 Preliminaries on Diffusion Models

In this section, we provide a brief overview of the concepts and techniques in denoising diffusion models that are necessary to understand our proposed method. Diffusion models are a class of generative models that are particularly effective at generating high-quality images (Sohl-Dickstein et al., 2015b; Nichol et al., 2021; Ramesh et al., 2022; Saharia et al., 2022; Rombach et al., 2022b). They aim to model a distribution $p_\theta(x_0)$ that approximates the data distribution $q(x_0)$ and is easy to sample from. DPMs model a "forward process" in the space of $x_0$ from data to noise by adding noise to real data, and a reverse process that tries to reconstruct the original data from the noisy version. The forward process is described by the equation

$$q(x_t|x_0) = \mathcal{N}(x_t; \sqrt{\bar{\alpha}_t}x_0, (1-\bar{\alpha}_t)\mathbf{I}),\tag{1}$$

where $x_{1:T}$ defines a set of noisy images and $x_0$ is the initial image. $\mathcal{N}$ denotes a Gaussian distribution, and $\bar{\alpha}_t$ are hyper-parameters. The reverse process is modeled by a Gaussian distribution

$$p_\theta(x_{t-1}|x_t) = \mathcal{N}(\mu_\theta(x_t), \Sigma_\theta(x_t)),\tag{2}$$

where neural networks are used to predict the mean and covariance of the distribution. The parameters of the model, $\theta$, are learned by optimizing a variational lower bound on the log-likelihood of the real data. Once trained, new images can be generated by starting from a noise sample and iteratively sampling from the reverse process distribution until reaching the final time step. In latent diffusion probabilistic models such as Stable Diffusion, these two processes are similar, while they proceeds in the latent space: $x_0$ is encoded into $z_0$ in an efficient, low-dimensional latent space first and then do the diffusion process. And in the case where a DPM is conditioned on additional information, such as text information $c$, the reverse process becomes $p_\theta(z_{t-1}|z_t, y)$, where $y$ is the input text.

## 4 Discriminative Diffusion Models

### 4.1 Problem Formulation

The problem of image-text matching is formalized as follows: given a text prompt $y \in \mathcal{Y}$ and a set of images $\mathcal{X}$, we aim to find the image $x^* \in \mathcal{X}$ that is most semantically aligned with the given text prompt $y$. Formally, we define the image-text matching problem as finding the function $f : \mathcal{Y} \times \mathcal{X} \to [0, 1]$ that assigns a score to each image-text pair $(y, x)$ indicating the degree of semantic alignment between the text and image. The goal is to find the image $x$ that maximizes the score for a given text prompt $y$, i.e., $x^* = \arg\max_{x \in \mathcal{X}} f(y, x)$.

## 4.2   Method Overview

To learn the function $f$, the main idea is to leverage the powerful representations learned by a pre-trained Stable Diffusion model to perform image-text matching. There are three key modules in Discffusion, *cross-attention score computation*, *LogSumExp pooling*, and *attention-based prompt learning*, as shown in Figure 2. The cross-attention score computation module extracts the mutual influence between visual and textual information by computing the attention scores from the cross-attention matrix in U-Nets of the Stable Diffusion model. The LogSumExp pooling module pools these attention scores over all tokens in the text description to obtain a single matching score. Finally, the attention-based prompt learning module fine-tunes the model by updating the key and value mappings from text to latent features in the cross-attention layers under a few-shot setting. This allows the model to learn new image-text concepts while retaining the ability to capture complex and nuanced relationships between images and text. The model outputs a score that measures the alignment between the image and text, which can be used to adapt the model from a text-to-image generation task to an image-text matching task. Note that Discffusion is a general pipeline that can adopt other faster sampling strategies as well.

## 4.3   Cross-attention Score Computation

Cross-attention scores can be a measure of the relevance of an image and a text to each other (Chen et al., 2020; Li et al., 2019). Prior research (Hertz et al., 2022b) shows that the cross-attention within diffusion models governs the layout of generated images and the scores in cross-attention maps represent the amount of information flows from a text token to a latent pixel. They are calculated by taking the dot product of the representations of the image and text in a latent space, and normalizing by the product of their norms. We propose to adapt cross-attention scores as a way to better capture the complex relationships between images and text in the image-text matching task. In the sequel, we elaborate on our strategy in depth.

Stable Diffusion (Rombach et al., 2022a) is trained to generate images from text prompts, and as such, it has learned strong compressed representations of both text and images. This enables us to use these representations to learn the function $f$ for image-text matching.

More specifically, a text prompt $y$ is first encoded into an intermediate representation $r_y = \tau_\theta(y) \in \mathbb{R}^{m \times d_\tau}$ using a domain-specific encoder $\tau_\theta$, where $m$ represents the number of text tokens. We then encode each image $x \in \mathcal{X}$ where $x \in \mathbb{R}^{H \times W \times 3}$ in RGB space into a latent image representation $z = \mathcal{E}(x)$, where $\mathcal{E}(x)$ is the image encoder. The noisy version of the input $z$ is obtained by:

$$z_t = \sqrt{\bar{\alpha}_t} z + \sqrt{1 - \bar{\alpha}_t} \epsilon \quad \text{for} \quad \epsilon \sim \mathcal{N}(\mathbf{0}, \mathbf{I}), \tag{3}$$

where constants $\bar{\alpha}_t$ are hyper-parameters inherited from Song et al. (2020), which will control the level of noises applied to the latent representation. The encoder $\epsilon_\theta$ in the U-Net (Ronneberger et al., 2015b) of the pre-trained text-to-image generation model then encode $z_t$ into $r_x = \varphi_i(z_t)$, where $\varphi_i(z_t) \in \mathbb{R}^{n \times d_\epsilon^i}$, $n$ and $i$ denote the number of image feature tokens and index of layers respectively (we dropped the superscript $i$ for $n$ for notation simplicity). This forms a (flattened) intermediate depiction within the U-Net that uses $\epsilon_\theta$, which are subsequently integrated into intermediate layers of the U-Net via a cross-attention mechanism defined as $\text{Attention}(Q, K, V) = \text{softmax}\left(\frac{QK^T}{\sqrt{d}}\right) \cdot V$, with $Q = r_x \cdot W^{q^{(i)}}, K = r_y \cdot W^{k^{(i)}}, V = r_y \cdot W^{v^{(i)}}$. Here, $W^{q^{(i)}} \in \mathbb{R}^{d_\epsilon^i \times d}, W^{k^{(i)}} \in \mathbb{R}^{d_\tau \times d}, W^{v^{(i)}} \in \mathbb{R}^{d_\tau \times d}$ are learnable projection matrices (Jaegle et al., 2021; Vaswani et al., 2017). These matrices map the inputs to query, key, and value matrices, respectively, where $d$ is the output dimension of the projection operations in the attention computation.

## 4.4   LogSumExp Pooling (LSE)

To compute the function $g$ and quantitatively evaluate the degree of semantic alignment between an image and a text prompt, we leverage LogSumExp (LSE) pooling (Blanchard et al., 2021) as a means of aggregating the attention maps generated by the cross-attention mechanism in our model. By using LSE pooling, we are able to take into account the relative importance of different image and text tokens in the attention map, rather than simply averaging or summing all elements in the map. This has several benefits. Firstly, LSE

pooling is able to handle large values and outliers in the attention map more robustly than other pooling methods, such as average or sum pooling. Secondly, LSE pooling has high numerical stability during training. Thirdly, LSE pooling is able to better preserve the ordering of values in the attention map, allowing for more interpretable and accurate matching scores.

For notation simplicity, we drop the batch and attention head dimension, the attention map matrix is denoted as $A \in \mathbb{R}^{n \times m}$, where $n$ represents the number of image tokens (height $\times$ width) in the latent space and $m$ represents the number of text tokens. To compute the image-text pair matching score, we utilize the LogSumExp (LSE) pooling operator, which is applied directly to each row of the attention matrix $A$, resulting in a vector of length $m$, and then averaged to produce the final score:

$$f(A) = \frac{1}{n} \sum_{i=1}^{n} \frac{1}{\lambda} \log \left( \sum_{j=1}^{m} \exp(\lambda A_{ij}) \right) \tag{4}$$

Here, $A_{i,:}$ represents the $i$-th row of the matrix $A$, Ave is the average operator, and $\lambda$ is a scaling factor that magnifies the importance of the most relevant pairs of image region features and attended text sentence vectors. By default, we set $\lambda = 1$. This operation computes the log-sum-exponential over each row, summarizing the contribution of each text token to the image features, and then averages these values across different timesteps to produce a single scalar score for the image-text pair $(y, x)$. We further enhance the model's capability for image-text matching through attention-based prompt learning, which we will introduce in the next section.

## 4.5 Attention-based Prompt Learning for Stable Diffusion

We aim to adapt the latent diffusion probabilistic model to the image-text matching task leveraging only a few examples, that is, under the few-shot setting. The task of fine-tuning aims at updating the mapping from the given text to the aligned image distribution, and the text features are only input to $W^k$ and $W^v$ projection matrix in the cross-attention block. Inspired by Liu et al. (2021); Lester et al. (2021); Li & Liang (2021), we propose the use of learnable prompts, which are added to the attention matrices in our model. Specifically, as shown in Figure 2, we introduce learnable prompt embedding matrices, which are added element-wise to the key and value attention matrices at the identified layer of the Stable Diffusion model. We keep the original $W^k$ and $W^v$ frozen, and only update the learnable prompt weights added to them. As our addition operation applies to all layers and sampled timesteps, we will omit superscripts $t$ and layer $l$ for notational clarity and obtains:

$$W^{k'} = W^k + p^k, W^{v'} = W^v + p^v. \tag{5}$$

Both $W^k$ and $W^v$ are frozen and $p^k$ and $p^v$ receive gradients and are updated during training, which is inspired by LoRA (Hu et al., 2021) and can further improve the training efficiency. This allows the model to adapt to new instances by attending to relevant information in the intermediate representation of the text inputs, $\tau_\theta(y)$. With the learned prompt embeddings in the few-shot scenario, we can effectively adapt the Stable Diffusion to image-text matching performance. The overall algorithm is shown in Algorithm 2. For optimization, we use the margin-based triplet loss function between the predicted match score and the true match score. Let $L$ be the loss, we have:

$$L = \sum_{i} \max \left(0, f(x_{\text{neg}}, y) - f(x_{\text{pos}}, y) + M\right), \tag{6}$$

where $f(\cdot, \cdot)$ represents the score function defined in Eq. (4) computed from the cross-attention map $A$ with image $x$ and text $y$, $i$ denotes the sample index, $x_{\text{pos}}$ is the groundtruth image corresponding to the $i$-th text $y$, $x_{\text{neg}}$ is the negative image, and $M$ is a predefined margin where we use 0.2 in our experiments.

**Dynamic Attention Head Weighting** We propose a method for adjusting the weights of different attention heads in the cross-attention of our model. We compute the gradient of the output of each attention head with respect to the input, and use this gradient to weight the contribution of each head to the final

---

**Algorithm 1** Discffusion Training

---

1: $x$: Image, $y$: Text
2: $\epsilon$: Noise
3: $z$: Latent representation, $z_t$: Noisy latent representation
4: $\mathcal{E}$: Encoder
5: $\tau$: Domain-specific encoder
6: $\varphi$: Intermediate representation of the U-Net
7: **for** $(x, y)$ in the batch **do**
8:     Image latent representation $z \leftarrow \mathcal{E}(x)$
9:     Noisy image latent representation $z_t \leftarrow z, \epsilon$            Eq. 3
10:     Text latent representation $r_y \leftarrow \tau(y)$
11:     Intermediate representation $r_x \leftarrow \varphi(z_t)$
12:     Compute attention maps $A \leftarrow r_y, r_x$
13:     Update attention maps with attributed attention maps $A \leftarrow \text{Attr}_h(A)$        Eq. 7
14:     Compute matching score $f(A)$                        Eq. 4
15:     Compute loss $L \leftarrow y$                            Eq. 6
16:     Update $W^{k'} \leftarrow W^k, W^{v'} \leftarrow W^v$          Eq. 5
17: **end for**

---

**Algorithm 2** Discffusion Inference

---

1: $x$: Image, $y$: Text
2: $\epsilon$: Noise
3: $z$: Latent representation, $z_t$: Noisy latent representation
4: $\mathcal{E}$: Encoder
5: $\tau$: Domain-specific encoder
6: $\varphi$: Intermediate representation of the U-Net
7: **for** $(x, y)$ in the batch **do**
8:     Image latent representation $z \leftarrow \mathcal{E}(x)$
9:     Noisy image latent representation $z_t \leftarrow z, \epsilon$            Eq. 3
10:     Text latent representation $r_y \leftarrow \tau(y)$
11:     Intermediate representation $r_x \leftarrow \varphi(z_t)$
12:     Compute attention maps $A \leftarrow r_y, r_x$
13:     Compute matching score $f(A)$                        Eq. 4
14: **end for**

---

output of the model as follows:

$$\text{Attr}_h(A) = A_h \odot \sum_{k=1}^{H} \frac{\partial f(A)}{\partial A_h}, \tag{7}$$

where $\odot$ is element-wise multiplication, $A_h \in \mathbb{R}^{n \times m}$ denotes the $h$-th head's attention weight matrix, and $\frac{\partial f(A)}{\partial A_h}$ computes the gradient along $A_h$. The $(i, j)$-th element of $\text{Attr}_h(A)$ computes the interaction in terms of the $h$-th attention head. The gradient weights are detached and utilized solely as scalar multipliers during training. During training, we then update the attention map with $\text{Attr}_h(A)$. By adjusting these weights, we are able to estimate the gradient flow, keep the most essential token alignment, guide the matching process between text and image and achieve better performance in our task. The inference time of Discffusion is introduced in the next section.

## 4.6 Inference

During inference, Discffusion leverages the latent representations of both images and text to compute a matching score that determines how well the image corresponds to the text prompt, as outlied in Algorithm 2. Specifically, for each pair of image $x$ and text $y$ in the batch, Discffusion first obtains the noisy latent representation $z_t$. Then, the text prompt $y$ is encoded into its latent representation $r_y$ using the domain-

specific encoder $\tau$. The U-Net is used to obtain the intermediate image representation $r_x$ from the noisy latent representation $z_t$. The attention maps $A$ are computed using the latent representations $r_y$ and $r_x$. Finally, the matching score $f(A)$ is computed from the attention maps from different sampling timesteps, where Discffusion can adopt various sampling strategies designed for diffusion-based generative models [1]. This process integrates both high-level and low-level features from the image and text during the diffusion process, effectively leveraging the strong representation of the diffusion model to perform image-text matching.

## 5 Experiments

### 5.1 Datasets

We use the Compositional Visual Genome (ComVG) (Krishna et al., 2017) and RefCOCOg (Yu et al., 2016) datasets to do image-text matching, which requires model's ability to understand fine-grained details, spatial relationships, and compositionality of image and text pairs. Additionally, we include the VQAv2 dataset, which we adapted for image-text matching by concatenating questions with their corresponding answers to form unique text prompts for matching with images, to demonstrate the versatility and robustness of Discffusion across different vision and language tasks. Apart from these, Winoground (Thrush et al., 2022) and VL-checklist (Zhao et al., 2022) are also included for evaluating the understanding ability of compositionality.

**Compositional Visual Genome (ComVG) (Krishna et al., 2017)** is a reconstructed dataset of the Visual Genome (Krishna et al., 2017) dataset, which contains 108,007 images annotated with 2.3 million relationships. These relationships are represented as subject-predicate-object triplets and include both action and spatial relationships. ComVG was created by selecting a subset of 542 images from Visual Genome that contain clear relationships, resulting in a total of 5400 data points. It mainly contains three subcategories: subject, predicate, and object. We perform evaluation on the entire dataset as well as on the 'Subject', 'Predicate', and 'Object' subcategories, respectively.

**RefCOCOg (Yu et al., 2016)** is a reconstructed dataset of the MS-COCO (Lin et al., 2014) dataset, including 85,474 referring expressions for 54,822 objects in 26,711 images, with a focus on images containing between 2 and 4 objects of the same category.

**VQAv2 (Goyal et al., 2017)** The VQAv2 dataset (Goyal et al., 2017) contains questions such as 'Binary', 'Other', 'Numbers', it is commonly converted to a classification task with 3,129 answer classes with frequency large than 9. In our setting, we modify the candidate text to be the concatenation of question and answer pair for each question and perform matching with images. We perform evaluation on all the modified dataset as well as on the 'Binary' and 'Other' subcategories, respectively.

### 5.2 Experimental Setup

We use Stable Diffusion v2.1-base with the xFormer (Lefaudeux et al., 2022) and flash attention (Dao et al., 2022) implementation, which utilizes the LAION (Schuhmann et al., 2022) dataset for pre-training. On the RefCOCOg dataset, we sample 10 text prompts from the pool each time, and the model is asked to do the correct matching given the image and the 10 sampled text prompts. We first evaluate our method under the zero-shot setting and select the variant with the best performance (using attention maps with dynamic attention head weighting, averaged across all U-Net layers, and using the LogSumExp, see the ablation studies section for details). We then test Discffusion under the setting where we train the model with only 5% of the dataset (Yoo et al., 2021), demonstrating its adaptation capability using limited data.

### 5.3 Baselines

We mainly compare Discffusion with several baseline models, including CLIP, BLIP2, and Diffusion Classifier. For CLIP, we use the CLIP-ViT/H-14 model with the OpenCLIP variant as the backbone for the fair comparison.

---

[1]By default in the paper, we use the DDIM (Song et al., 2020) sampling strategy.

Table 1: Comparison of accuracy (%) under the fine-tuning setting with 5% training data and zero-shot setting (Average of three runs). Discffusion can beat CLIP and Diffusion Classifier by a large margin consistently across all these datasets under the fine-tuning setting, demonstrating the superiority of our approach compared with traditional vision and language models pre-trained for discriminative tasks. Note that OFA's pre-training datasets include Visual Genome (VG), RefCOCOg, and VQAv2, and BLIP2's pre-training datasets also include Visual Genome (VG) and COCO (Lin et al., 2014) (whose images are the sources images for RefCOCOg and VQAv2). Therefore, for a fair comparison, CLIP should be the main baseline to evaluate the effectiveness of Discffusion.

| | Method | Compositional Visual Genome | | | | RefCOCOg | | VQAv2 | | |
|---|---|---|---|---|---|---|---|---|---|---|
| | | Subjects | Objects | Predicate | All | Top-1 Acc. | Top-5 Acc. | Binary | Other | All |
| Zero-shot | CLIP | 78.87 | 80.59 | 58.60 | 74.20 | 68.10 | 82.79 | 66.31 | 4.90 | 17.55 |
| | BLIP2* | 80.96 | 84.07 | 64.14 | 80.65 | 75.28 | 89.20 | 67.73 | 6.73 | 20.49 |
| | OFA* | 80.79 | 82.49 | 60.89 | 76.28 | 72.05 | 88.52 | 66.01 | 6.37 | 19.57 |
| | Diffusion Classifier | 79.85 | 82.41 | 63.28 | 77.05 | 73.02 | 88.60 | 67.10 | 5.95 | 19.64 |
| | Discffusion | 80.62 | 84.74 | 63.27 | 78.11 | 73.07 | 89.16 | 67.51 | 6.82 | 20.43 |
| Fine-tuned | CLIP (Fine-tuning) | 80.77 | 82.49 | 60.50 | 76.10 | 69.88 | 84.57 | 66.94 | 5.10 | 17.86 |
| | CLIP (Prompt Learning) | 78.88 | 79.51 | 60.41 | 74.24 | 69.40 | 84.48 | 67.32 | 5.16 | 18.03 |
| | BLIP2* | **81.12** | 84.23 | 64.30 | 80.81 | 76.43 | 90.35 | **71.23** | 7.23 | **20.99** |
| | OFA* | 80.80 | 82.50 | 60.90 | 76.29 | 73.31 | 89.78 | 66.45 | 6.81 | 20.01 |
| | Diffusion Classifier | 80.48 | 82.52 | 63.42 | 77.50 | 73.40 | 89.83 | 68.94 | 6.78 | 19.88 |
| | Discffusion | 80.78 | 84.90 | 63.43 | 78.27 | 75.87 | 89.96 | 70.60 | 6.91 | 20.52 |
| | Discffusion (w/ pre-training) | 80.98 | **84.97** | **65.17** | **80.89** | **76.91** | **90.42** | 71.04 | **7.25** | 20.83 |

- CLIP (Fine-tuning) (Radford et al., 2021): We fine-tune the CLIP model, adjusting only the last layer.

- CLIP (Prompt Learning) (Zhou et al., 2022): Inspired by prompt learning strategies, we incorporate learnable prompts to textual inputs conditioned on individual images.

- BLIP2 (Li et al., 2023b): A vision-language pre-training model that bridges the modality gap using a lightweight Querying Transformer.

- Diffusion Classifier (Li et al., 2023a): Extracts standard classifiers from class-conditional diffusion models by choosing the conditioning that best predicts the noise added to the input image.

## 5.4 Experiments: Comparison under the Fine-tuned Setting

In order to facilitate a fair comparison, we adapt the use of the Discffusion model to two distinct settings on ComVG, given the different resolutions of Stable Diffusion and CLIP. The first setting involves resizing all images in the dataset to a resolution of 224x224 first, and then upsampling and resizing to 512x512 for the use of Discffusion, so that Discffusion will not take advantage of its higher input resolution. The results are shown in Table 1. The other setting involves directly resizing all images to 224x224 and 512x512 base resolution, which is employed by CLIP and Stable Diffusion, with the results shown in the Appendix.

We present a comprehensive comparison with other state-of-the-art methods: BLIP2 (Li et al., 2023b) and OFA (base) (Wang et al., 2022), specifically under the few-shot scenario using only 5% of the data in Table 1. Our results show that our method is competitive, trailing BLIP2 closely, which is the state-of-the-art across numerous vision and language tasks. It's worth noting that OFA's pre-training datasets include Visual Genome (VG), RefCOCOg, and VQAv2, and BLIP2's pre-training datasets also include Visual Genome (VG) and COCO (Lin et al., 2014) (whose images are the sources images for RefCOCOg and VQAv2). Therefore, for a fair comparison, CLIP should be the baseline to evaluate the effectiveness of our approach for few-shot learning. As shown in Table 1, Discffusion can beat CLIP consistently across all these datasets. In addition, with pre-training on the corresponding datasets, Discffusion outperforms all other methods, including the state-of-the-art BLIP2 on Compositional Visual Genome, RefCOCOg, and the 'Other' category on VQAv2.

Table 2: Comparison of accuracy (%) on Winoground and VL-checklist across three runs for CLIP, BLIP2, and Discffusion, both zero-shot and fine-tuned on MS-COCO. For BLIP2's evaluation on Winoground, it is approached as an image-text matching task to ensure a fair comparison, rather than prompting the model to answer the rewritten question from captions (Yarom et al., 2024). BLIP2 gains little from fine-tuning on MS-COCO because BLIP2's pre-training datasets already include MS-COCO (Lin et al., 2014), and Winoground is quite different from MS-COCO as it requires compositional understanding ability. Similarly, Diffusion Classifier, designed as a zero-shot approach, does not benefit much from fine-tuning. As can be seen, under the fine-tuned setting, Discffusion outperforms all other approaches by a large margin across both datasets in each category. The detailed results on Winoground with 95% confidence intervals across different categories are presented in Table 10.

| | Method | Winoground | | | VL-checklist | | | |
|---|---|---|---|---|---|---|---|---|
| | | Text | Image | Group | Attribute | Object | Relation | Average |
| Zero-shot | CLIP | 30.68 | 11.91 | 8.36 | 67.82 | 75.67 | 67.11 | 70.18 |
| | BLIP2 | 31.78 | 12.37 | 9.02 | **79.00** | **84.05** | **73.55** | **78.86** |
| | Diffusion Classifier | **37.75** | 12.25 | 9.00 | 64.19 | 75.98 | 66.99 | 68.23 |
| | Discffusion (Ours) | 34.11 | **13.38** | **11.04** | 73.12 | 79.58 | 69.48 | 74.06 |
| MS-COCO Fine-tuned | CLIP | 31.50 | 12.00 | 9.25 | 67.90 | 75.75 | 68.15 | 70.73 |
| | BLIP2 | 28.25 | 13.00 | 9.09 | 79.64 | 84.70 | 73.95 | 79.97 |
| | Diffusion Classifier | 26.76 | 10.00 | 8.75 | 65.90 | 76.81 | 67.14 | 68.94 |
| | Discffusion (Ours) | **35.75** | **14.50** | **12.75** | **79.80** | **85.90** | **74.58** | **80.52** |

## 5.5 Experiments: Zero-shot Transfer from Training on the MS-COCO Dataset

We assess the zero-shot transfer capabilities of the Discffusion model and its comparison with notable baselines such as CLIP, BLIP2, Diffusion Classifier (Li et al., 2023a). We fine-tuned all these models on the MS-COCO dataset, and then performed the evaluation on downstream datasets. Given that BLIP2's pre-training datasets include Visual Genome (VG) and COCO (Lin et al., 2014), and considering the overlap of MS-COCO images with those in ComVG and RefCOCOg, we selected two commonly used evaluation benchmarks: Winoground (Thrush et al., 2022) and VL-checklist (Zhao et al., 2022). The evaluation results are depicted in Table 2. Discffusion outperforms all others on Winoground and VL-checklist, showcasing its superior ability to generalize the learned visual-linguistic representations to datasets beyond the ones used during fine-tuning.

## 5.6 Experiments: Zero-shot Evaluation on the Winoground and VL-checklist Dataset

We conducted additional experiments to evaluate the potential of Discffusion for zero-shot inference with 50 sampling steps on the Winoground (Thrush et al., 2022) and VL-checklist datasets, which require compositional understanding ability. The top of Table 2 shows a comparison of Discffusion with other zero-shot approaches. Discffusion outperforms CLIP and is competitive with Diffusion Classifier under the zero-shot setting on both the two datasets, even though Discffusion is primarily designed as a fine-tuned method. The detailed performance across categories <relation, object, both> in the format (text score, image score, group score) is shown in Table 10 in the Appendix. As shown, Discffusion excels notably in the 'relation' category, demonstrating proficiency in discerning shuffled elements like verbs, adjectives, prepositions, and adverbs. This suggests that Discffusion, with its generative-task-oriented pre-training and our adaptation strategy, offers enhanced capability for handling fine-grained nuances, spatial interrelationships, and intricate compositionalities compared to conventional discriminative vision and language models.

## 5.7 Discussion about Inference Efficiency of Discffusion

The inference memory cost of Discffusion is approximately 9.8GB when using the batch size of one. We use DDIM sampling to get the sampling timesteps. The inference speed is heavily dependent on the number of sampling timesteps. We demonstrate the trade-off between the number of sampling timesteps and the

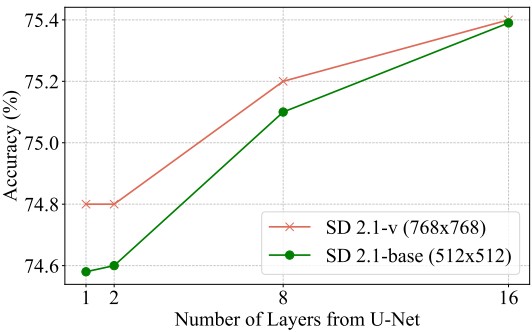

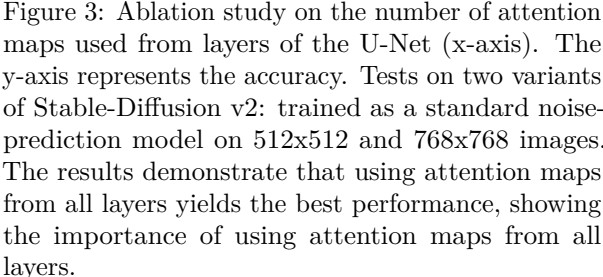

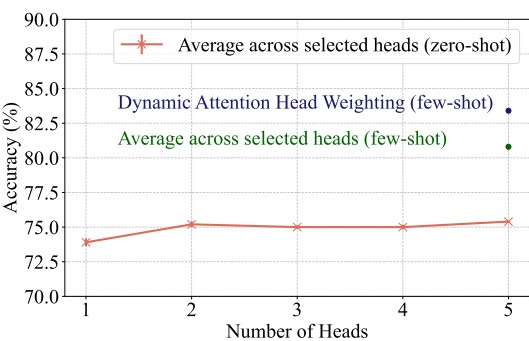

Figure 3: Ablation study on the number of attention maps used from layers of the U-Net (x-axis). The y-axis represents the accuracy. Tests on two variants of Stable-Diffusion v2: trained as a standard noise-prediction model on 512x512 and 768x768 images. The results demonstrate that using attention maps from all layers yields the best performance, showing the importance of using attention maps from all layers.

Figure 4: Ablation study on the number of attention heads (five in total within the Stable Diffusion) in the U-Net (x-axis) with few-shot performance on the ComVG dataset (y-axis) under the two scenarios: using the average of all attention maps and using our dynamic attention head weighting method. The results illustrate the superiority of our weighting method.

Table 3: Ablation study on the ComVG dataset across three runs using cosine similarity, maximum value from each column of the attention map, and the smoothed maximum (LogSumExp pooling); and the amount of noise added during the diffusion process: using consistent noise levels of 0.4, 0.8, and using ensembling. The accuracy numbers are in (%).

|  | Method/Noise Level | Subjects | Objects | Predicate | All |
|---|---|---|---|---|---|
| Method | Cosine similarity | 50.7 | 42.5 | 30.5 | 43.0 |
|  | Maximum (column-wise) | 58.8 | 59.5 | 40.4 | 57.7 |
|  | Maximum (row-wise) | 57.8 | 59.0 | 40.1 | 57.0 |
|  | LogSumExp | 78.9 | 80.1 | 61.2 | 75.4 |
| Noise Level | Noise level 0.4 | 75.5 | 77.5 | 58.4 | 72.5 |
|  | Noise level 0.8 | 75.7 | 77.7 | 58.5 | 72.7 |
|  | Ensembling | 76.1 | 78.0 | 59.2 | 73.1 |

inference time on the RefCOCOg dataset with a batch size of 16. The inference was executed in a distributed manner on a NVIDIA workstation equipped with 4 A6000 GPUs, leveraging the capabilities of the Accelerate library [2] as shown in Figure 5. We observe that the Top-1 Accuracy tends to stabilize when the number of sampling timesteps approaches approximately 100.

## 5.8 Ablation Studies

In this section, we delve deeper into the nuances of our experimental findings in a zero-shot setting on the ComVG dataset. ComVG's complexity allows us to examine the finer-grained cross-modal alignment at the entity level, showcasing how each design aspect of our method impacts the compositional matching of disentangled concepts.

**Effect of Attention Maps from Different Sets of U-Net Layers** This study examines the effect of different layer configurations in the U-Net architecture of Stable Diffusion v2 on the computation of attention maps. We use two variants of Stable Diffusion v2 and take the average of the attention maps from different

---

[2]https://huggingface.co/docs/accelerate/index

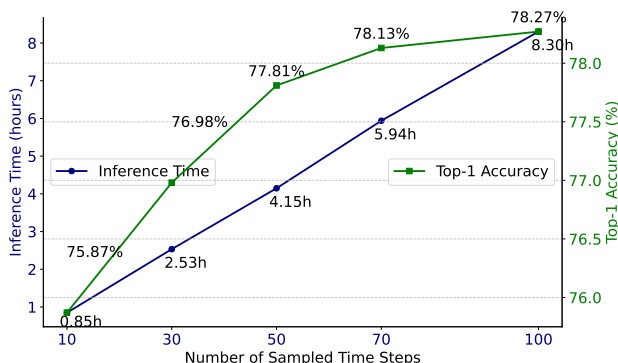

Figure 5: Inference time and accuracy versus sampling steps on RefCOCOg with Discffusion, conducted on a 4 GPU setup.

layers of U-Net. Following insights from Kwon et al. (2022); Tang et al. (2022)—which suggest that deeper layers capture more semantically relevant information, we consider the last one, the last two, the last eight, and all layers. The results, shown in Figure 3, demonstrate that incorporating all layers yields the highest accuracy, indicate that a comprehensive integration of both high-level and low-level features is crucial for tasks like image-text matching, maintaining both coarse and detailed aspects of image representation. Interestingly, focusing solely on the last two layers also results in performance closely mirroring that of using just the final layer, indicating a potential redundancy in the task-specific information contained within these later stages.

**Cosine Similarity *vs.* Maximum *vs.* LogSumExp Pooling for Score Computation**  We compare the overall accuracy of using Cosine Similarity, Maximum value from the attention map, and LogSumExp Pooling for score computation in Table 3. For cosine similarity, the dimensionality of the latent representations for images and texts differs. Therefore, we employ projection layers to project the representations from both modalities into a common dimensionality. For the maximum value method, we evaluate two variants: (i) extracting the maximum value from each column of the attention map (column-wise), and (ii) extracting the maximum value from each row of the attention map (row-wise). The results indicate that LogSumExp Pooling achieves the highest accuracy, followed by the maximum value method, with Cosine Similarity yielding the lowest performance. This may be because LogSumExp can smooth out the influence of individual noisy elements, resulting in more robust and accurate matching scores. Additionally, LogSumExp can effectively capture the overall importance of each element in the attention map, rather than relying solely on the maximum value. Overall, these results highlight the effectiveness of LogSumExp as a method for computing the matching score in image-text matching tasks.

**Effect of Dynamic Attention Head Weighting**  We investigate the effect of using varying numbers of attention heads in Stable Diffusion on image-text matching performance, specifically, the attention heads in the cross-attention layer of the spatial transformer. To do this, we randomly sample a subset of heads and leave the rest unused. The results, as shown in Figure 4, indicate that using all heads (five in total in our evaluated model version) from the Spatial Transformer (Jaderberg et al., 2015) located in the U-Net of Stable Diffusion performs the best. Additionally, we compare the results of using dynamic attention head weighting versus averaging across selected heads (few-shot) to evaluate the impact of attention head selection on performance, and dynamic attention head weighting performs better.

**Ensembling over Noise Levels**  In diffusion models, the level of noise controls the variance of the Gaussian noise added during the diffusion process, which can affect the degree of change in the generated image. Our study, illustrated in Table 3, explores the effects of varying noise levels in the diffusion process. To further improve the performance of our method, we use an ensemble technique inspired by Wolleb et al. (2022) by averaging the scores over four relative different noise levels: {0.2, 0.4, 0.6, 0.8}. This is done by first obtaining the score under each noise level scenario and then averaging them. The results of this comparison

are shown in Table 3. Our experimental results demonstrate that this ensemble technique leads to a noticeable improvement in performance.

## 6 Conclusion

In this paper, we proposed a method for matching text and images in the latent space of Stable Diffusion. We fine-tuned the U-Net part of the model by focusing on the cross-attention between text features and latent image features, which reflects the alignment between text and image. Our results show that this fine-tuning approach improves the alignment between text and image, leading to better performance in image-text matching tasks. Overall, our approach is a pioneering work that leverages the inherent flexibility of diffusion-based visual generative models, demonstrating improved explainability and an enhanced ability to capture fine-grained details, spatial relationships, and compositionality. Our results can motivate research on simpler alternatives to adapt Stable Diffusion models, as well as on future methods for better utilization of them.

### Broader Impact Statement

The Discriminative Stable Diffusion (Discffusion) approach proposed in this paper is dependent on a pre-trained Stable Diffusion model, which may be challenging to obtain in certain scenarios where the model has yet to be publicly released or where the computational resources required for training are not available. Additionally, the quality of the pre-training can greatly impact the performance of Discffusion, highlighting the need for further research to investigate methods for improving the pre-training process. There are other techniques such as multi-task learning and meta-learning that can be possibly incorporated to improve the performance of Discffusion. It is worth noting that in real-world scenarios, there is often a limited amount of labeled data available, and collecting more data can be costly and time-consuming. Therefore, the ability of Discffusion to perform well under a few-shot setting is an important aspect of its potential utility in practical applications.

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

## A    Ablations on Input Images Resolutions

In our experimental setup, CLIP accepts image resolutions of 224x224 and Stable Diffusion accepts 512x512. Considering that image resolutions within the ComVG dataset vary, we adopted two distinct resizing strategies to maintain a balanced comparison. In the first approach, we initially resize all input images to a 224x224 resolution, followed by a subsequent resizing to 512x512 specifically for Discffusion. In this way, there is no additional information introduced when comparing with using CLIP. In the second approach, we directly resize the input image resolutions to 512x512 and 224x224 for Discffusion and CLIP respectively. The results of these strategies are illustrated in Table 4. It is evident that Discffusion surpasses both the fine-tuned CLIP and the CLIP with prompt learning baselines by a large margin. These outcomes suggest that the Discffusion model can adeptly harness the benefits of larger input image resolutions.

## B    Training Efficiency of Discffusion

The training efficiency of the Discriminative Stable Diffusion (Discffusion) model is noteworthy. Remarkably, it requires only a single NVIDIA V100 GPU for training, with a memory footprint of approximately 15.4 GB with the Accelerate library [3] and FlashAttention (Dao et al., 2022) using the implementation available in HuggingFace Diffusers [4]. The model demonstrates efficient training times across different datasets: it completes training on the Compositional Visual Genome (ComVG) dataset in 7 hours, on the RefCOCOg dataset in 11 hours, and on the more extensive Visual Question Answering version 2 (VQAv2) dataset in 17 hours. These training durations highlight the model's practicality and feasibility for use in various research and application scenarios, especially considering the moderate computational resources required.

## C    Adapting Discffusion to Latent Consistency Models (Luo et al., 2023)

This section investigates the adaptability of Discriminative Stable Diffusion (Discffusion) to Latent Consistency Models (LCMs) — a class of faster diffusion models that require fewer steps for both training and inference (four steps). We evaluate the performance of Discffusion extended with LCMs in two experimental setups: training on the MS-COCO dataset followed by zero-shot testing on Winoground, and few-shot training on ComVG with subsequent testing on the same.

The empirical outcomes, detailed in Table 5, reveal that Discffusion when augmented with LCMs surpasses the original Discffusion on Winoground. However, it underperforms on the ComVG dataset. Closer examination of the training dynamics for Discffusion (LCMs) on ComVG indicates instability, as depicted by the fluctuating loss curve in Figure 6. This instability may due to the small number of sampling steps and account for the diminished results on ComVG. Nonetheless, the preliminary success on Winoground suggests that, with further refinement and larger dataset fine-tuning, LCMs could enhance the capabilities of Discffusion. This finding demonstrates the potential of Discffusion to excel when integrated with more advanced diffusion models, paving the way for future research in this direction.

## D    Visualization of Learned Prompt Weights

We present a visualization of the prompt weights learned after training on the MS-COCO dataset in Figure 7. The distribution of weights is predominantly near zero, showcasing sparsity in the update made to the original Stable Diffusion weights. This sparsity suggests that only minor updates are required for Stable Diffusion to adapt to discriminative tasks effectively, which underscores the model's potential for such applications and strong latent representations gained during pre-training.

---

[3]https://huggingface.co/docs/accelerate/index
[4]https://huggingface.co/docs/diffusers/index

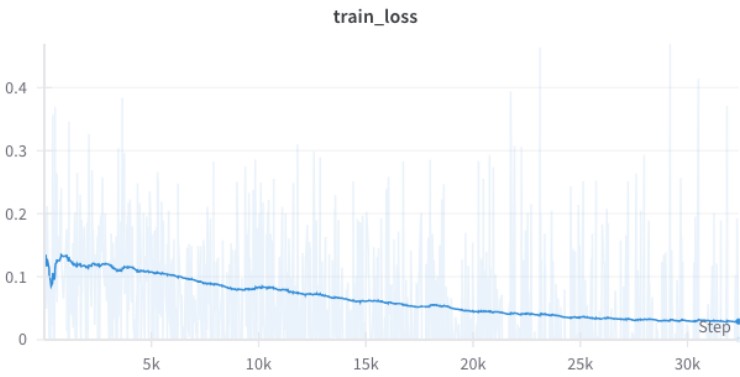

Figure 6: Training loss curve for Discffusion with LCMs on the ComVG dataset. The observed fluctuations point to training instability, which potentially impacts performance on the few-shot setting.

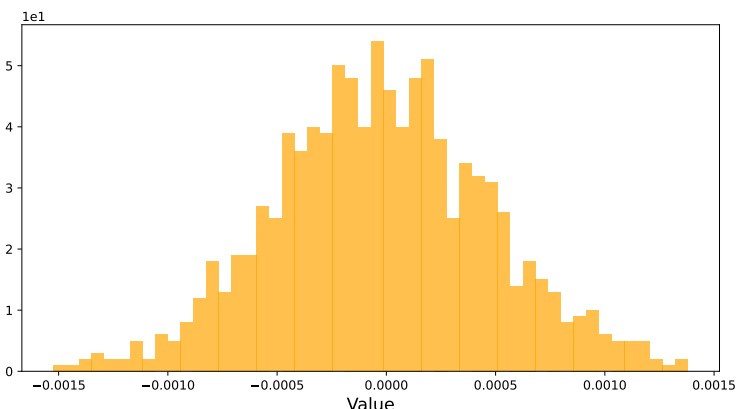

Figure 7: Visualization of learned prompt weights after training on MS-COCO.

# E   Image-text Retrieval Directly Using Discffusion Embeddings

In this section, we study the quality of the representation space learned in Discffusion. We directly using Discffusion's embedding to do retrieval. Specifically, after fine-tuning Discffusion on the Compositional Visual Genome (ComVG) and RefCOCOG dataset, we use the latent space features after the VAE to do the retrieval. This process involves using cosine similarity to match these features with corresponding text embeddings. We then compare the performance of Discffusion's embeddings with those from the CLIP model. The results, as shown in Table 6, illustrate the capability of Discffusion embeddings for retrieval tasks, underscoring their potential for future research and applications in this area.

# F   Additional Experimental Results

## F.1   Extreme Few-shot Setting

We have expanded our experimentation to extreme few-shot learning by conducting tests with only 0.5% of training data (27 examples from ComVG), with the results shown in Table 7. These results illustrate the benefits of ours with such small data, particularly when compared to CLIP's performance.

Table 4: Comparison of accuracy (%) on Compositional Visual Genome (ComVG) using CLIP, Discriminative Stable Diffusion under the few-shot setting with different input image resolutions.

| Method | Resolution | Compositional Visual Genome | | | |
| --- | --- | --- | --- | --- | --- |
| | | Subjects | Objects | Predicate | Average |
| CLIP (Fine-tuning) | 224x224 | 80.77 | 82.49 | 60.50 | 76.10 |
| CLIP (Prompt Learning) | 224x224 | 78.88 | 79.51 | 60.41 | 74.24 |
| Discffusion | 224x224 | 80.81 | 83.17 | 63.51 | 78.11 |
| Discffusion | 512x512 | **80.97** | **84.32** | **63.53** | **78.19** |

Table 5: Comparison of accuracy (%) on Winoground and VL-checklist of CLIP, BLIP2, and Discffusion fine-tuned on MS-COCO.

| Method | Winoground | | | ComVG | | | |
| --- | --- | --- | --- | --- | --- | --- | --- |
| | Text | Image | Group | Subjects | Objects | Predicate | Average |
| Discffusion | 34.00 | 13.75 | 10.50 | **80.78** | **84.90** | **63.43** | **78.27** |
| Discffusion (LCM) | **34.75** | **15.75** | **11.00** | 79.84 | 83.33 | 60.08 | 73.47 |

## F.2 Masking Out Objects in RefCOCOg

We conduct additional experiments by masking objects with the provided bounding boxes to observe their impact on the RefCOCOg dataset using 300 images. When elements are masked, the Top-1 accuracy of Discriminative Stable Diffusion (Discffusion) drops significantly from 81.7% to 43.6%, as shown in Table 8. This substantial decrease in accuracy demonstrates that Discffusion can capture key objects in images for the correct matching.

## F.3 Ablation Study on Time Steps Sampling in Discffusion

As presented in Table 9, we explored the effectiveness of using cross-attention maps from different time steps in the Discriminative Stable Diffusion (Discffusion) compared to variants that employ a single time step's cross-attention maps (the sampled first time step and the sampled last time step). The comparative results demonstrate the enhanced performance of the original Discffusion method. Leveraging all time steps' information can effectively integrate and process the coarse-to-fine information throughout the diffusion steps to do matching.

## F.4 Ablation Study on Loss Functions in Discffusion

Vision and language models trained with contrastive objective may not preserve all the visual information. In this section, we study the impact of using different loss functions in training Discffusion on image-text matching performance on the ComVG dataset. We test the cross-entropy loss: $L_{CE} = -\frac{1}{N} \sum_{n=1}^{N} (y_n \log \hat{y}_n + (1 - y_n) \log (1 - \hat{y}_n))$ where $N$ is the number of samples in the training dataset and $y_n$ and $\hat{y}_n$ are the true and predicted match scores for the $n$-th sample, respectively; and the standard denoising diffusion training objective: $L_{Diffusion} = \mathbb{E}_{x,\epsilon,t} \left[ \| \epsilon - \epsilon_\theta (x, t) \|^2 \right]$. The results, as presented in Table 11, indicate that Discffusion, employing the original margin-based triplet loss function outperforms the alternative approaches. This superiority is primarily attributable to the original diffusion training loss's inefficacy in adapting Stable Diffusion for discriminative tasks and the relative inadequacy of the cross-entropy loss in managing the nuances of image-text matching tasks.

Table 6: Accuracy comparison (%) between directly using CLIP-embedding and Discffusion-embedding to do retrieval on the Compositional Visual Genome (ComVG) and RefCOCOg datasets.

| Method | Compositional Visual Genome | | | | RefCOCOg | |
|---|---|---|---|---|---|---|
| | Subjects | Objects | Predicate | Average | Top-1 Acc. | Top-5 Acc. |
| CLIP-embedding | 80.77 | 82.49 | 60.50 | 76.10 | 69.88 | 84.57 |
| Discffusion-embedding | 78.43 | 76.25 | 57.68 | 73.27 | 66.14 | 84.61 |

Table 7: Comparison of Few-shot and Extreme Few-shot Learning in Compositional Visual Genome.

| Method | Few-shot (ComVG) | | | | Extreme Few-shot (ComVG) | | | |
|---|---|---|---|---|---|---|---|---|
| | Subjects | Objects | Predicate | Average | Subjects | Objects | Predicate | Average |
| CLIP (Fine-tuning) | 80.77 | 82.49 | 60.50 | 76.10 | 74.40 | 75.10 | 57.90 | 69.80 |
| CLIP (Prompt Learning) | 78.88 | 79.51 | 60.41 | 74.24 | 74.55 | 75.15 | 57.79 | 69.83 |
| Discffusion (Ours) | **80.78** | **84.90** | **63.43** | **78.27** | **78.65** | **81.40** | **66.68** | **75.91** |

### F.5 Additional Zero-shot results on the Winoground dataset

In Table 10, we show additional results on the Winoground dataset in accuracy with 95% confidence intervals across the three different categories. Discffusion excels notably in the 'relation' category, demonstrating proficiency in discerning shuffled elements like verbs, adjectives, prepositions, and adverbs.

## G Results on Text-to-Image Generation

In addition to the existing evaluations in the main paper, we also evaluate the capability of fine-tuned Discffusion model on the task of text-to-image generation. We use two scores to evaluate the quality of generated images:

- Frechet Inception Distance (FID) (Heusel et al., 2017) measures the similarity between two sets of images. It considers both the quality and diversity of the generated images. It calculates the distance between the distributions of the real and generated images in the feature space of a pre-trained Inception model. A lower FID score suggests a higher similarity to the real image dataset, implying better image quality and variety.

- Inception Score (IS) (Barratt & Sharma, 2018), on the other hand, was proposed to evaluate the quality and diversity of images generated by GANs. It leverages the Inception model, originally used for image classification, to score generated images. The Inception Score is calculated based on the assumption that good models should generate diverse images (high entropy) but still be able to classify these images correctly (low conditional entropy). Hence, a higher IS indicates better performance of the generative model.

We detail the performance of these metrics on both the MS-COCO (Lin et al., 2014) and Compositional Visual Genome (ComVG) datasets in Table 12. The results show that the Discriminative Stable Diffusion (Discffusion) model, when fine-tuned in a few-shot setting for discriminative tasks on the ComVG dataset does not compromise the generative capability of the model, which remains on par with the performance of the original Stable Diffusion model.

Table 8: Comparison of Top-1 and Top-5 accuracy (%) on RefCOCOg using Discriminative Stable Diffusion (Discffusion) under the few-shot setting, and Discriminative Stable Diffusion (Discffusion) with the impact of masking elements. This table highlights the effect of element masking on the Discffusion approach in the RefCOCOg dataset.

| Method | Top-1 Acc. | Top-5 Acc. |
|---|---|---|
| Discffusion | **81.74** | **91.96** |
| Discffusion (Masking Elements) | 43.60 | 54.78 |

Table 9: Comparison of accuracy (%) on Compositional Visual Genome (ComVG). The table shows the effectiveness of utilizing cross-attention maps from different time steps compared to using only single time step's cross-attention map.

| Method | Subjects | Objects | Predicate | Average |
|---|---|---|---|---|
| Discffusion | **80.78** | **84.90** | **63.43** | **78.27** |
| Discffusion (First Time Step Sampling) | 59.93 | 54.24 | 51.68 | 55.45 |
| Discffusion (Last Time Step Sampling) | 76.78 | 80.90 | 59.43 | 74.27 |

## H Qualitative Results

### H.1 Visualization of Generate Samples with Discffusion

In this section, we study the generative abilities after adapting Stable Diffusion to a discriminative one. We present visualization of generated samples from the Visual Genome dataset with both Discffusion and Stable Diffusion in Figure 8. The image quality of Discffusion is comparable to Stable Diffusion.

### H.2 Visualization of Cross-attention Maps

We present visualizations of averaged cross-attention maps prior to the integration of the learned prompt in Figure 9. These representations elucidate the Discffusion model's reasoning strategy for ground-truth image-text pairs. During the diffusion process, the correspondence between the text and image is reflected in token-level scores on the attention map. Tokens receiving higher scores become predominant in the decision-making process for matching. It is clear from the visualizations that the regions of the attention maps imbued with informative elements tend to correspond with higher scores. The higher attention score implies a stronger similarity between a token in the given text and elements of the image.

## I Error Analysis

Here, we present an example of a failure case on the Compositional Visual Genome (ComVG) dataset, as illustrated in Figure 10, where the Discffusion model failed to successfully discriminate. As can be observed, Discffusion struggles to recognize the word '`white`' in the image, leading to incorrect matching. This could potentially be attributed to the lack of ability to understand compositionality inherent in the original stable diffusion model.

## J Hyper-parameter Settings

In this section, we will introduce the detailed hyper-parameter settings in the paper.

Datasets: We used the Compositional Visual Genome and RefCOCOg datasets for the image-text matching task, where we used the split introduced in Jiang et al. (2022) and Yu et al. (2016).

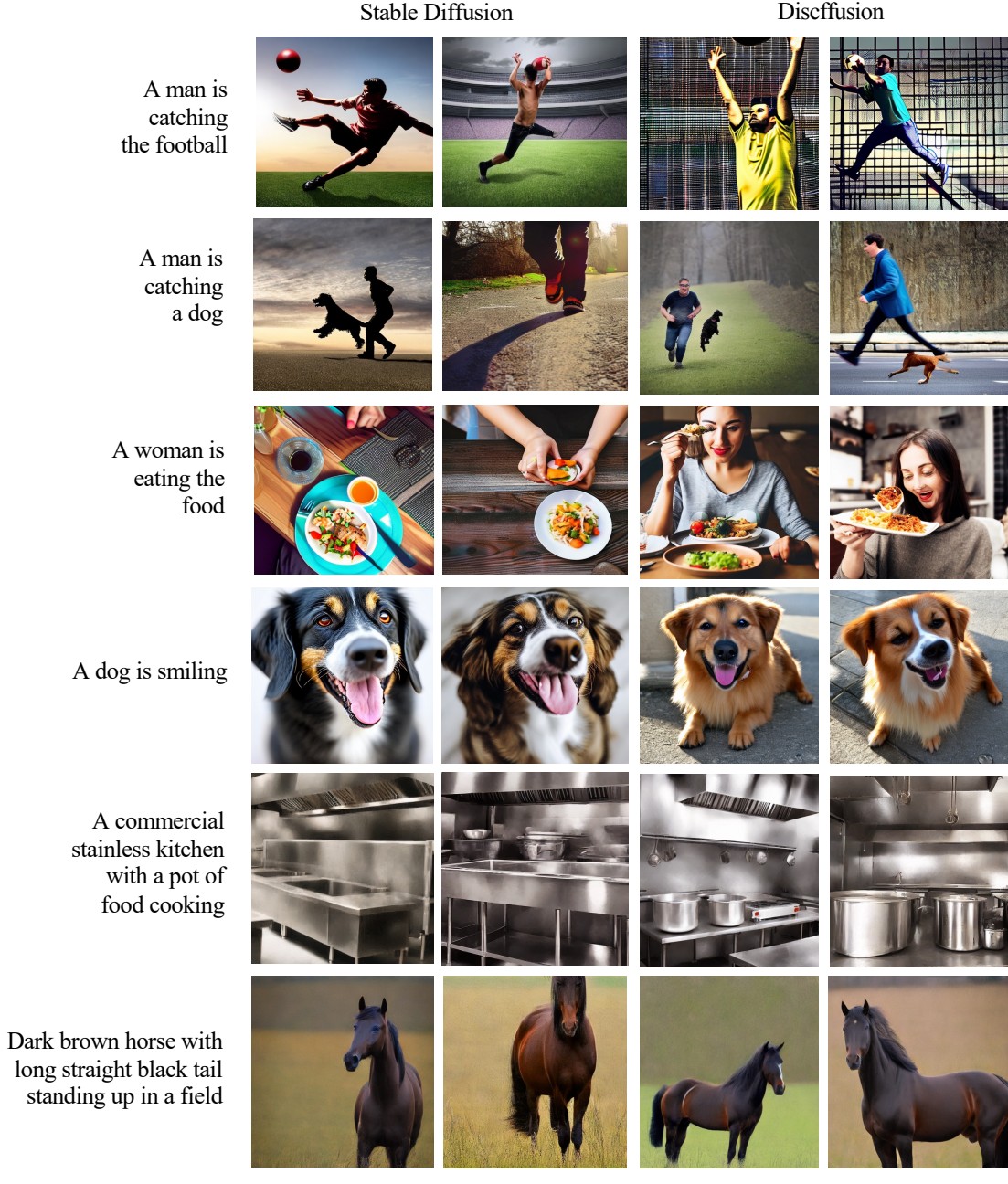

Figure 8: The visualizations of generated samples from the Visual Genome dataset using the prompts `A man is catching the football`, `A man is catching a dog`, `A woman is eating the food`, `A dog is smiling`, `A commercial stainless kitchen with a pot of food cooking`, and `Dark brown horse with long straight black tail standing up in a field`, employing a DDIM sampler with 50 steps. The **left two columns** are samples generated by the original Stable Diffusion. The **right two columns** are samples generated by Discffusion. The image quality of Discffusion is comparable to Stable Diffusion, suggesting that the few-shot fine-tuning on discriminative tasks in Discffusion does not hinder text-to-image generation performance.

Table 10: Zero-shot results on the Winoground dataset in accuracy with 95% confidence intervals. Discffusion excels notably in the 'relation' category, demonstrating proficiency in discerning shuffled elements like verbs, adjectives, prepositions, and adverbs.

| Category | Relation | | | Object | | | Both | | |
|---|---|---|---|---|---|---|---|---|---|
| | **Text** | **Image** | **Group** | **Text** | **Image** | **Group** | **Text** | **Image** | **Group** |
| CLIP | $0.22_{\pm 0.02}$ | $0.08_{\pm 0.02}$ | $0.05_{\pm 0.02}$ | $0.36_{\pm 0.01}$ | $0.12_{\pm 0.02}$ | $0.08_{\pm 0.02}$ | $\mathbf{0.81}_{\pm 0.03}$ | $0.45_{\pm 0.03}$ | $\mathbf{0.42}_{\pm 0.04}$ |
| BLIP2 | $0.22_{\pm 0.01}$ | $0.05_{\pm 0.01}$ | $0.04_{\pm 0.01}$ | $0.35_{\pm 0.01}$ | $\mathbf{0.20}_{\pm 0.01}$ | $\mathbf{0.13}_{\pm 0.01}$ | $0.62_{\pm 0.04}$ | $0.37_{\pm 0.02}$ | $0.37_{\pm 0.03}$ |
| Discffusion | $\mathbf{0.25}_{\pm 0.01}$ | $\mathbf{0.10}_{\pm 0.02}$ | $\mathbf{0.07}_{\pm 0.03}$ | $\mathbf{0.40}_{\pm 0.01}$ | $0.13_{\pm 0.01}$ | $0.12_{\pm 0.02}$ | $0.80_{\pm 0.02}$ | $\mathbf{0.46}_{\pm 0.02}$ | $\mathbf{0.42}_{\pm 0.03}$ |

Table 11: Comparison of accuracy (%) on Compositional Visual Genome (ComVG) trained with different loss functions.

| Method | Subjects | Objects | Predicate | Average |
|---|---|---|---|---|
| Discffusion | **80.78** | **84.90** | **63.43** | **78.27** |
| Discffusion (Cross-Entropy Loss) | 75.77 | 81.42 | 60.05 | 72.01 |
| Discffusion (Diffusion Loss) | 65.58 | 59.83 | 54.42 | 60.96 |

Model architecture: We used the Stable Diffusion v2.1 model with a U-Net architecture and a VAE, with resolutions of 768x768 and 512x512, and CLIP-ViT/H as the text encoder. The sampling was carried out using the DDIM (Song et al., 2020) method with a total of 50 steps.

Pre-training: The model was pre-trained from scratch using the LAION-5B (Schuhmann et al., 2022) dataset. The U-Net architecture consists of 16 layers with input blocks, middle blocks, and output blocks. The input blocks contain a 6-layer spatial transformer, the middle block contains a 1-layer spatial transformer, and the output block contains a 9-layer spatial transformer. Each spatial transformer contains two attention layers, one self-attention layer and one cross-attention layer.

Evaluation metrics: We evaluated the performance of our approach using the Acc@$K$ metric, with $K=\{1,5\}$, measuring the accuracy of the correct match of image-text pairs.

Ablations: In the ablation study on the number of attention heads used, we experimented with dynamic attention head weighting. To obtain the weights, we first trained the model and used the learned weights in the subsequent inference time. For the other two settings, where we averaged the attention head weights, we used consistently averaged weights across all the experiments. This allowed us to compare the effect of the number of attention heads on the final performance of the model, while controlling for the potential confounding factor of different attention head weights; In the ablation study of using cosine similarity, as the dimension of the latent representation of images and texts are not the same, we compute the cosine similarity on the shared dimensions between the image and text representation. For the experiments of using the maximum value, we take the maximum value from each column of the attention map; In the ablation study on performing ensembling over noise levels, we set the noise level to {0.2, 0.4, 0.6, 0.8} and obtain the prediction under each scenario. We then average the predictions to obtain the final score.

Table 12: Comparison of Frechet Inception Distance (FID) and Inception Score (IS) for Text-Conditional Image Synthesis on the RefCOCOg and Compositional Visual Genome (ComVG) datasets. We use DDIM sampler and the total number of sampling steps is 50.

| Method | Text-Conditional Image Synthesis | | | |
|---|---|---|---|---|
| | MS-COCO | | ComVG | |
| | FID ↓ | IS ↑ | FID ↓ | IS ↑ |
| Stable Diffusion | 23.3 | 19.7 | 26.9 | 17.5 |
| Discffusion | 24.1 | 19.5 | 26.0 | 18.1 |

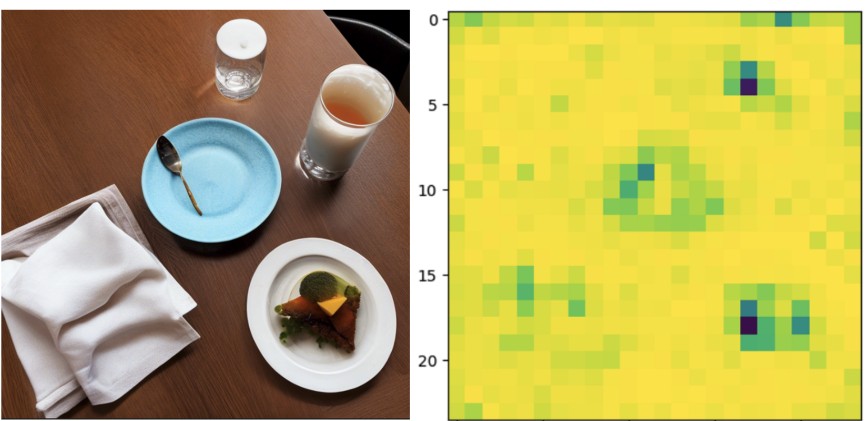

Figure 9: The visualization of averaged attention maps in Discffusion for the prompt: `A table with a plate of food, a glass of water, and a napkin.`

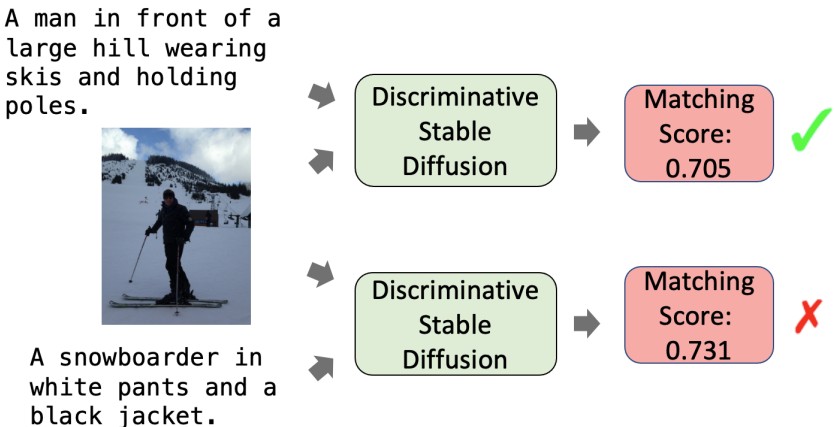

Figure 10: The failure case in Discffusion for the prompt: `A man in front of a large hill wearing skis and holding poles.`

