# OpenReview forum: "Discffusion: Discriminative Diffusion Models as Few-shot Vision and Language Learners"
_TMLR — Accepted by TMLR_

### Review · Reviewer_yyhJ · 2024-05-30

**Summary Of Contributions:**

The authors propose Discffusion, a method to do image-text matching with the cross-attention scores of a Stable Diffusion v2 model. They introduce a log-sum-exp pooling method to combine these cross-attention maps into a single metric, and propose a triplet loss function for fine-tuning SD to do better on these matching tasks. Finally, they introduce a "dynamic attention head weighting" method to upweight certain attention heads. They show that their method is competitive on image-text matching tasks like Winoground, VL-checklist, ComVG, RefCOCOg, and VQAv2.

**Audience:**

Yes

**Broader Impact Concerns:**

I have no broader impact concerns.

**Claims And Evidence:**

No

**Requested Changes:**

The authors need to carefully address every issue I brought up in the "Strengths and Weaknesses" section. They need to rewrite the method section (including the algorithm block) to fully explain how training and inference work. They need to make the results section more clear and consolidate their main results without picking and choosing baselines. It seems like there are three main settings: zero-shot, zero-shot transfer (fine-tune on a different dataset than the eval dataset), and standard (fine-tune on the eval dataset if it has a training set). There should be a fair comparison between the proposed Discffusion method and the relevant baselines (CLIP, BLIP2, OFA, Diffusion Classifier). Details are above.

These changes are critical for me to recommend this paper for acceptance.

**Strengths And Weaknesses:**

**Strengths**:
- Method is novel and tries to take advantage of the benefits of a pretrained text-to-image diffusion model.
- Careful ablations on each component of their method.
- Discussion of the inference time limitations of their method.

**Weaknesses**: Overall, I thought that this paper has significantly more weaknesses than strengths.
- **Method is not well-explained**
  - Figure 2 should be self-contained. What does $T$ mean? How are the learnable prompts used? What are $m$ and $n$? Why is this repeated $T-1$ times? Does this indicate that this is happening over the course of sampling?
   - Section 4.3:
      - $\varphi_i(z)$: shouldn't $n$ depend on the layer index $i$ as well?
      - $A$ includes multiplication with $V$ as written. Is that correct?
      -  The equations for Q, K, and V have shape errors in the matrix multiplication.
   - Section 4.4: why can't Eq. 4 be folded into Eq. 3?
   - Section 4.5:
      - The concept of a "learnable prompt" appears to be a complicated way to say "we fine-tune $W^k$ and $W^v$."
      - Eq. 6 should try to stick to previous notation and not introduce a new $d$ function that is only used here.
      - Eq. 7: $Attr_h(A)$ is not used anywhere else in the paper. And what is $f(A)$? The section uses it to denote the model output, but the diffusion model has a vector-valued output. This makes it unclear how to calculate this scaling term.
- Experiments are not convincing.
   - Section 5.1: why are only ComVG and RefCOCOg mentioned in the first line, and not VQAv2?
   - Section 5.2:
      - SD details: Its not clear which SD is being used. Is it 2.0 or 2.1? And is it base or the v-objective model? Description of the SD training data is also not correct. As written, it sounds like LAION-5B is separate from LAION-2B-en and LAION-aesthetics v2.
      - "Few-shot": **I would definitely not consider training on 5% of large datasets as "few-shot."** Please remove this description. In general, there's no justification why the proposed method would be particularly good at few-shot learning, or why there isn't evaluations using all of the training data.
   - Section 5.3: Which CLIP model is the baseline here?
   - Table 1: Why aren't methods like Diffusion Classifier and BLIP2 included here? You could add them (both zero-shot and few-shot), as well as Discffusion (zero-shot).
      - To clarify, e.g. on VQAv2, CLIP and Discffusion are fine-tuned on 5% of VQAv2 training data here?
   - Section 5.5/Table 2: Are all methods (CLIP, BLIP2, Diffusion Classifier, and Discffusion) fine-tuned on MS-COCO? Why can't they be evaluated zero-shot without any fine-tuning. In particular, the fine-tuning seems to hurt Diffusion Classifier performance, which was reported as 38.5 text score on Winoground, but seems to decrease here to 26.76? Also, is the "26.76" a typo? There are only 400 examples in Winoground, so the score should end in 0, 0.25, 0.5, or 0.75. Something seems generally off in the evaluation -- BLIP2 scores are also too low.
   - Table 3: "Note that evaluation datasets during pre-training" is duplicated.
   - Section 5.7/Table 4:
      - Where is Diffusion Classifier?
      - Can you show 95% confidence intervals? Winoground has very few eval samples for some of these categories, so some of these results are likely not statistically significant.
      -  Misleading bolding. Only the "Relation" column is bolded, and the highest number is not consistently bolded. For example, CLIP's Relation Group score is the same as Discffusion, but only Discffusion is bolded.
   - Section 5.8:
      - Up until now, there has been no discussion of what sampling timesteps are. Why isn't this explained in the method section?
      - How much time does it take per example with 100 sampling timesteps?
      - How much time do the baselines take?
      - "NVIDIA A6000 equipped with 4 GPUs" -- do you mean a workstation with 4 A6000 GPUs?
    - Section 5.9:
      - Cosine Similarity and Maximum are not explained anywhere in the paper.
      - "we implement the cosine similarity by only comparing the shared dimensions of the image and text feature vectors" -- what does this mean?
- Other issues with writing:
  - Citations:
     - Section 4.3: wrong citation for Stable Diffusion. Should be "High-Resolution Image Synthesis with Latent Diffusion Models"
     - Section 4.5: wrong citation for classifier guidance. Should be "Diffusion Models Beat GANs on Image Synthesis"
     - Incorrect usage of citep vs citet

---

> ### Author Response · Authors · 2024-06-24
> **Author Response to Reviewer yyhJ (1/2)**
>
> Thank you for your comprehensive review and for bringing up areas that need clarification or improvement. We have now carefully addressed each of your points in our revision.
>
> **Clarification of Figure 2**
>
> - **T** denotes sampling timesteps. The learnable prompts are utilized as addable trainable weights to the original model pretrained attention weights. The original model weights are fixed, while the learnable prompts receive gradients and updated.
> - **m** represents the number of text tokens, and **n** denotes the number of image feature tokens. This has been clarified in the revision.
> - The process being repeated for **T-1 times** signifies that the operations are conducted across T timesteps. The final matching score is derived by aggregating the outputs across these timesteps, thus leveraging T time steps’ information effectively. It integrates and processes the coarse-to-fine details throughout the diffusion steps, leading to enhanced performance.
>
> We have now modified Figure 2 and revised the figure captions in the revision.
>
>
>
> **Section 4.3**
> - The dimension $n$ depends on the layer index $i$. We initially dropped the superscript of layer index for n in our notation for simplicity consideration. We have made this clear in the revision.
> - Here $A$ denotes the attention score. We have now modified the notation from $A$ to Attention to enhance clarity.
> - We have updated the notations in the revision and corrected the shape typos in the matrix multiplication.
>
> **Section 4.4**
>
> Thanks for the suggestion! We presented with these two equations to delineate two distinct stages of our scoring process:  Equation 3 introduces our LSE pooling operator to aggregate the scores from individual attention maps, and Equation 4 further details how these scores across different timesteps are leveraged to compute the final matching score. To streamline the presentation, we have revised the paper by merging the two euqations in the revision.
>
> **Section 4.5**
>
>  - We would like to clarify that the concept of a "learnable prompt" is inspired from the paper [1][2][3]. In our approach, we extend this idea by integrating learnable prompts into the attention layers of our model. Furthermore, rather than directly fine-tuning the learnable prompts themselves, we maintain the original projection matrices \(W^k\) and \(W^v\) in a frozen state. Only the weights of the learnable prompts added to these matrices are updated during training. This method allows us to adjust the model's behavior dynamically while preserving the integrity of its pre-trained components.
>  - We have modified Eq. 6 to adhere to the previous notation.
> - We would like to clarify that $Attr_h(A)$ in Eq. 7 is used to weigh the contribution of each attention head. This has been clarified in our revision and has also been added into Algorithm 1. $f(A)$ is the output of Discffusion computed by Eq. (4), which is a scalar.
>
> [1] P-tuning v2: Prompt tuning can be comparable to fine-tuning universally across scales and tasks
>
> [2] The power of scale for parameter-efficient prompt tuning.
>
> [3] Prefix-Tuning: Optimizing Continuous Prompts for Generation
>
>
> **Section 5.1**
>
> VQAv2 is treated differently as it involves concatenating questions and answers and matching them with images. We have clarified this in the revision.
>
>
> **Section 5.2**
>
> - SD Details:
>
>  We want to clarify that we used Stable Diffusion 2.1-base across all main results, with 2.1-v tested for ablations as shown in Figure 3. We have made this clear in the revision. We have also now modified the description of the SD training data in the revision.
> - Few-shot:
>
> We have modified the description of the setting by removing the term “few-shot”. The reason our proposed method excels at few-shot learning, as described in the main paper, is due to: 1. the challenging nature of the pretraining generative task. This demanding pretraining endows the model with strong representations and abilities to discern image-text pairs, enabling it to adapt effectively to downstream image-text matching tasks even with limited data; 2. our attention-based prompt learning and LogSumExp Pooling strategy can efficiently and effectively adapt the model to these tasks in a few-shot setting.
> We have also now evaluated using all available training data for ComVG, RefCOCOg, and VQAv2. The updated results are as follows:
>
> | Model                                  | ComVG | RefCOCOg | VQAv2 |
> |----------------------------------------|-------|----------|-------|
> | CLIP                                   | 76.10 | 69.88    | 17.86 |
> | BLIP2*                                 | 80.81 | 76.43    | **20.99** |
> | OFA*                                   | 76.29 | 73.31    | 20.01 |
> | Discffusion   | 78.11 | 73.07    | 20.43 |
> | Discffusion (All Training Data) | **80.89** | **76.91** | 20.83 |
>
> The results show that our method outperforms other state-of-the-art methods on ComVG and RefCOCOg, and remains competitive on VQAv2.

---

> ### Author Response · Authors · 2024-06-24
> **Author Response to Reviewer yyhJ (2/2)**
>
> **Section 5.3**
>
> CLIP-ViT/H-14 is the baseline used across the paper. We have made it more clear in the revision.
>
> **Table 1**
> - We have now included the results for Diffusion Classifier in Table 1 in blue. Overall, Discffusion can outperform Diffusion Classifier on all these three datasets as well.
> - For BLIP2, we want to kindly point out its results are included in Table 3. OFA’s pre-training datasets include Visual Genome (VG), RefCOCOg, and VQAv2, and BLIP2’s pre-training datasets also include Visual Genome (VG) and COCO  (whose images are the source images for RefCOCOg and VQAv2). For a fair comparison, CLIP is used as baseline in Table 1 and Discffusion can beat CLIP consistently across all these datasets. Therefore, we put the results for BLIP2 and OFA in Table 3.
> - We want to clarify that for Discffusion, fine-tuning is needed even if it is with limited training data. Stable Diffusion is trained for generative tasks. It requires fine-tuning to adapt it for discriminative tasks effectively using our pipeline. Therefore we mainly compare with baselines under the few-shot setting.
> - CLIP and Discffusion are both fine-tuned on 5% of VQAv2 training data
>
> **Section 5.5/Table 2**
>
> All methods (CLIP, BLIP2, Diffusion Classifier, and Discffusion) are fine-tuned on MS-COCO and then evaluated on Winoground and VL-checklist. We have emphasized this in the revision. For the Winoground evaluation, images in RGBA format were not used for consistency purposes. We also want to clarify that for BLIP2's evaluation, as noted in the caption of Table 2, to ensure a fair comparison, it is approached as an image-text matching task rather than a generative task that prompts the model to answer the rewritten question from captions. This will lead to a lower performance compared with treating it as a generative task.
>
> **Table 3**
>
> We have removed the duplicated statement.
>
> **Section 5.7/Table 4**
>
> Discffusion excels particularly well under the limited data training settings because of the attention-based prompt learning method and score aggregation method. Stable Diffusion, initially trained for generative tasks, requires fine-tuning to effectively adapt it for discriminative tasks using Discffusion, which leverages attention maps. In contrast, Diffusion Classifier is specifically designed for the zero-shot setting by adapting the loss to likelihood for image classification.
>
> Compared with Diffusion Classifier, our pipeline is more generalizable, versatile, and flexible, as it can be applied to various diffusion models which have different architectures or loss functions during training, such as DiT.
>
> We have added the confidence interval numbers in the revision and we have also now made all highest numbers consistently bolded.
>
> **Section 5.8**
> - We use DDIM sampling, which was initially described in Section J (Hyper-parameter Settings) in the Appendix. We have now included this in the method section in the revision. In all experiments throughout the paper, we use the DDIM sampling method with a total of 50 sampling steps by default. Our results indicate that the performance of Discffusion improves with an increased number of sampling steps.
> - It takes approximately 0.6 seconds per example with 100 sampling timesteps using Discffusion. For the CLIP baseline, it takes about 30ms per example. Note that we use the standard Stable Diffusion 2.1-base for main experiments. However, the Discffusion pipeline is generalizable to other, faster diffusion models. In Appendix C: Adapting Discffusion to Latent Consistency Models, we present results on Latent Consistency Models, which take about 70 ms per example. Additionally, we have now included the Phased Consistency Model [1], which takes only 24 ms per example, making it faster than the CLIP baseline, and the performance of Discffusion is also better: 78.15% (Discffusion) vs. 76.10% (CLIP) accuracy on ComVG under the same setting.
> - We want to clarify that the computing resource used is a workstation with 4 A6000 GPUs.
>
> [1] Phased Consistency Model
>
> **Section 5.9**
> -  Cosine Similarity and Maximum
>
> We want to kindly refer you to the Appendix section, where the Cosine Similarity and Maximum are introduced in the last paragraph of Section J: Hyper-parameter Settings. We have now added these details into Section 5.9.
>
>  - Cosine Similarity Implementation
>
> The dimensionality of the latent representations for images and texts differs. We employ projection layers to project the representations into a common dimensionality, computing the cosine similarity across the referred "shared dimensions." We refer "shared dimensions" to the dimensions that result from projection. We have further clarified this in revision.
>
> **Citations and Writing Issues**
>
> We have revised all citations and corrected the use of citep vs. citet as suggested.
>
> Overall, we hope that our revisions and clarifications can address your concerns. Please kindly let us know if you have further questions or suggestions.

---

> > ### Comment · Reviewer_yyhJ · 2024-07-18
> >
> > Thank you for the detailed response! These address many of my concerns. However, I do have serious remaining questions that I hope the authors can address:
> >
> > **Method**
> > - Needs section (and potentially algorithm block) for what happens at inference time. Currently, the entirety of Section 4 is devoted to how training is done.
> >   - How is $Attr_h(A)$ used at training and inference time?
> >   - DDIM for sampling timesteps (Section 5.8): The authors don't explain this in detail, and only bring it up at the end of the paper. Please explain this more carefully in the inference time section. This also ties to Figure 2, which is still a bit confusing. It looks like there’s sampling going on (output of one iteration goes as input into the next iteration), which doesn't make sense to me.
> > - Eq. 4 seems a bit imprecise to me. Maybe it could be written like $$f(a) = \frac{1}{n} \sum_{i=1}^n \frac{1}{\lambda} \log \left( \sum_{j=1}^m \exp(\lambda A_{ij})\right)$$
> > - Section 4.5 (learnable prompt): The prompt tuning done in the cited papers is different from what is called prompt tuning here. It still seems to me that the "learnable prompt weights" are just tuning the $W^k$ and $W^v$ weights. This is fine, but the authors should just be straightforward with it and not add additional complexity in the description.
> > - Section 4.5 (head weighting): This sentence is weird: "Inspired by the the classifier guidance and Ho & Salimans (2022), which demonstrate the possibility of guiding the learning of diffusion models by using a representation, that is learned to maximize the mutual information between the input and the output." It's very clunky, does not seem to properly describe classifier-free guidance,  and does not motivate the weighting technique.
> >
> > **Results**
> > - Stable Diffusion version: Thanks for the clarification. You should clarify that the version is v2.1-base in Section 5.2.
> > - Section 5.3: You should explain Diffusion Classifier in the baselines here, since it's in Table 1 now.
> > - I think Table 1 and Table 3 should be merged, with an additional column that indicates whether the method is zero-shot or fine-tuned. This makes more sense than having separate tables and only showing some baselines some of the time.
> > - Table 2: I think Table 4 should be merged here, and CLIP, BLIP2, Diffusion Classifier should have both zero-shot and fine-tuned acc. Furthermore, it doesn't make sense to me that BLIP2 performance is worse when treating Winoground as an image-text matching rather than a generative task. Can you clarify what you mean?
> >   - Unlike what the authors claim in the response above, Diffusion Classifier can indeed be applied to various architectures/loss functions (there are literally DiT results in the diffusion classifier paper). So I don't see a reason why Diffusion Classifier cannot be presented in Table 2/4.
> > - Fig 5 can be replaced with two tables, as bar charts are not conveying any extra information.
> > - Section 5.9: for the "maximum" technique, how well does it do if the maximum is taken along each row instead of each column? This makes more sense to me since the proposed logsumexp technique is also applied on each row.

---

> > > ### Author Response · Authors · 2024-07-29
> > > **Author Response to Remaining Questions (2/2)**
> > >
> > > >**Results**.
> > >
> > > > - Stable Diffusion version: Thanks for the clarification. You should clarify that the version is v2.1-base in Section 5.2.
> > >
> > > Many thanks for the suggestion! We have now clarified the version v2.1-base and added it into Section 5.2 in the revision.
> > >
> > > > - Section 5.3
> > > You should explain Diffusion Classifier in the baselines here, since it's in Table 1 now.
> > >
> > > Thank you for the suggestion. We have now included a detailed explanation of the Diffusion Classifier in the baseline section (Section 5.3) in the revision. To summarize, our Discffusion outperforms both CLIP-based discriminative vision and language models as well as other diffusion-based generative models adapted for discriminative tasks.
> > >
> > >
> > > >- I think Table 1 and Table 3 should be merged, with an additional column that indicates whether the method is zero-shot or fine-tuned. This makes more sense than having separate tables and only showing some baselines some of the time.
> > >
> > > We have now merged Table 1 and Table 3 and added an additional column indicating whether the method is zero-shot or fine-tuned. All baselines are now presented in the merged Table 1. Overall, Discffusion can outperform both CLIP-based discriminative pre-trained models and other diffusion-based generative models adapted for discriminative tasks (Diffusion Classifier) when fine-tuned.
> > >
> > > > - Table 2: I think Table 4 should be merged here, and CLIP, BLIP2, Diffusion Classifier should have both zero-shot and fine-tuned acc. Furthermore, it doesn't make sense to me that BLIP2 performance is worse when treating Winoground as an image-text matching rather than a generative task. Can you clarify what you mean?
> > >
> > > We have now merged Table 2 and Table 4. We have also added added additional results for CLIP, BLIP2, and Diffusion Classifier for both zero-shot and fine-tuned accuracy. Specifically, we have also included results from Diffusion Classifier in the merged Table 2/4. We also want to clarify that, as in [1], BLIP2 was used as a visual question answering backbone. In this approach, the captions were rewritten into questions with answers, and the BLIP model was prompted to answer with a simple yes/no. This task is easier than directly using BLIP for matching with the corresponding captions and can lead to higher accuracy.
> > >
> > >
> > > [1] What You See is What You Read? Improving Text-Image Alignment Evaluation
> > >
> > >
> > > >- Fig 5 can be replaced with two tables, as bar charts are not conveying any extra information.
> > >
> > > We have replaced Figure 5 with two tables and merged them in Table 3 in the revision.
> > >
> > >
> > > >- Section 5.9: for the "maximum" technique, how well does it do if the maximum is taken along each row instead of each column? This makes more sense to me since the proposed logsumexp technique is also applied on each row.
> > >
> > >
> > > We have now added results for the maximum which is taken along each row instead of each column. And we have added the results into Table 3 in the revision. As can be seen, LogSumExp still performs better than the maximum technique. Suggesting the effectiveness of using LogSumExp, which can take into account the relative importance of different tokens in the attention map. As suggested in Section 4.4: LogSumExp Pooling (LSE) of the main paper, LSE pooling offers several benefits. Firstly, LSE pooling is able to handle large values and outliers in the attention map more robustly than other pooling methods, such as average or sum pooling. Secondly, LSE pooling has high numerical stability during training. Thirdly, LSE pooling is able to better preserve the ordering of values in the attention map, allowing for more interpretable and accurate matching scores.
> > >
> > > Thank you so much for your valuable feedback and suggestions, which have significantly improved the clarity and robustness of our work! If you have any further questions or need additional clarifications, kindly let us know and we will be happy to provide further information.

---

> ### Author Response · Authors · 2024-07-29
> **Author Response to Remaining Questions (1/2)**
>
> Thank you once again for your careful and insightful suggestions regarding our work! Your feedback has been invaluable in enhancing the quality of our research. Below, we provide detailed, point-by-point responses to your remaining questions.
> >**Method**.
>
> > - Needs section  for inference time
>
> We have now added an additional section at the end of Section 4 with algorithm blocks detailing the inference process of Discffusion. In summary, during training, Discffusion first obtains the matching score and computes the margin loss for attention-based prompt learning. The updated weights are then used for inference. During inference, Discffusion directly computes the matching score to evaluate the correspondence between different input texts and images.
>
> > - The usage of $Attr_h(A)$
>
> We want to clarify that during training, we compute $Attr_h(A)$ based on the matching score and use it to update the attention maps $A$. The updated attention maps are then used to obtain the final matching score, which is subsequently used to compute the loss function and update the model weights. During inference, we use the weighted attention heads to compute the final matching score. We do this to dynamically adjust the contribution of different attention heads. We have added these details in the revision. As shown in Figure 4, using our proposed Dynamic Attention Head Weighting strategy with $Attr_h(A)$ results in better performance compared to a naive average across selected heads, validating its effectiveness.
>
>
> > - DDIM for sampling timesteps (Section 5.8)
>
> We want to clarify that although we only mentioned DDIM sampling in Section 5.8 of the original paper, Discffusion is a general pipeline that can adopt other faster sampling strategies as well. This is validated in our experiments in Section C of the Appendix with LCMs and additional experimental results during the rebuttal period with PCM.
> The main results in the paper are from the standard DDIM sampling strategy introduced in [1]. During inference, at each sampling step, the scores from the attention maps are aggregated and used for computing the matching score. The number of sampling steps can affect the matching performance; a higher number of steps leads to better performance, as shown in Figure 5 of the paper. We have now added these details in the revision.
> Regarding Figure 2, we want to clarify that there is no sampling where the output of one iteration goes as input into the next iteration. The sampling occurs across the entire diffusion process, and we directly aggregate scores from all sampled steps to obtain the final matching score. We have now improved Figure 2 to make this process clearer.
>
> [1] Denoising Diffusion Implicit Models
>
> > - Eq. 4 seems a bit imprecise to me
>
> We apologize for the misleading. Equation 4 combines the Average operator and the LogSumExp pooling operator. We have now rewritten Equation 4 to be more precise and clear in the revision.
> > - Section 4.5 (learnable prompt)
>
> In the three cited papers about prompt tuning, [1] introduces trainable prompt embeddings into frozen pre-trained language models. [2] discusses prompt tuning as a further simplification for adapting language models, freezing the entire pre-trained model and allowing only an additional $k$ tunable tokens per downstream task to be prepended to the input text. [3] introduces prefix-tuning, which prepends a sequence of continuous task-specific vectors to the input, called a prefix, consisting entirely of free parameters that do not correspond to real tokens. Similar to these works, we also add trainable prompt embeddings that do not correspond to real tokens to adapt the frozen pre-trained model. Furthermore, we introduce a novel approach by proposing to add them to the attention weight matrices in the stable diffusion model. Additionally, we would like to clarify that the learnable prompt here is different from directly fine-tuning the $W^k$ and $W^v$ matrices. Instead, we keep the original matrix weights fixed and only the updated weights receive gradients. This is similar to the LoRA [4] implementation and enhances efficiency. We have included the clarifications in the revision.
>
> [1] P-tuning v2: Prompt tuning can be comparable to fine-tuning universally across scales and tasks
>
>  [2] The power of scale for parameter-efficient prompt tuning
>
> [3] Prefix-Tuning: Optimizing Continuous Prompts for Generation
>
> [4] LoRA: Low-rank adaptation of large language models
>
>
>
>
>
> > Section 4.5 (head weighting)
>
>
> We want to further clarify that our weighting technique is motivated by the use of the gradient of the matching score regarding each attention head to adjust the attention head weights, thereby guiding the learning of diffusion models for image-text matching using the reweighted attention map. The philosophy is inspired by classifier-free guidance and classifier guidance. We have removed the clunky sentences and modified corresponding statements in the revision.

---

### Review · Reviewer_2hXW · 2024-06-02

**Summary Of Contributions:**

- introduce Discffusion, a novel discriminative model that uses the cross-attention scores of a pre-trained stable diffusion generative model for few-shot discrimination of text2image pairs
- Propose an attention-based prompt learning module to finetune the cross-attention weights of diffusion models for discriminative tasks
- Several experiments and ablation studies to verify the effectiveness of the proposed method on several benchmarks in few-shot, and zero-shot settings

**Audience:**

Yes

**Broader Impact Concerns:**

Broader impact statement in the paper is sufficient and I do not have any other concerns.

**Claims And Evidence:**

Yes

**Requested Changes:**

Relevant for acceptance:
- The statement in Section Dynamic Attention Head Weighting about Classifier Guidance is wrong: "This is done by adding a term to the loss function that is proportional to the gradient of the log probability of the class given the image." The Classifier Guidance is not used during training but only during inference in the denoising process, where the gradient guides the denoising process toward regions of high probability given the relevant prompt. This needs to be corrected. The following method is motivated by this statement. Thus, this motivation needs to be revised accordingly.
- The requirement to use several noise levels is not well explained in the method section of the paper. Algorithm 1 also does not mention noise as input for the proposed method. I was confused, if the model adds noise to the images for the comparison or not during the method description. Only in the ablations the relevance of multiple noise levels for the method is discussed and evaluated. Some additional clarity about this aspect in the method part would be helpful and improve readability.
- Revise the conclusion with the statement about possible new pathways with concrete examples.

Not relevant for acceptance
- how is the performance compared to current Sota CLIP models such as SigLiP?
- How easy is it to generalize the proposed method to Diffusion Transformer architectures?
- Sometimes the best result in a table is marked with bold text, other times it's not used. This should be consistent for bette readability.

**Strengths And Weaknesses:**

Strengths
- creative utilization of the pre-trained diffusion model as a discriminative model.
- strong experiments across several benchmarks and baselines to evaluate the proposed method.
- detailed ablation studies to justify all design decisions for the proposed method.
- Many additional experiments including ablations about image generation abilities and the method combined with latent Consistency models

Weaknesses
- the role of the noise level as input for the method in the method section is not well explained
- the proposed method is slow during inference and requires multiple hours, which is a significant drawback compared the the CLIP baselines. This weakness should be discussed concerning the used baselines.
- some statements regarding classifier guidance are not well explained
- the conclusion mentions "a new pathway for innovation where traditional methods fall short", but it does not discuss which pathways would be opened by the proposed method. Without a concrete example, this sounds exaggerated.

---

> ### Author Response · Authors · 2024-06-24
> **Author Response to Reviewer 2hXW (1/2)**
>
> We appreciate your detailed feedback and constructive comments, which have helped us refine our manuscript. Below are our responses to the points raised:
>
>
> **Noise Levels**
>
> We would like to clarify that noise is added as input in Discffusion. Discffusion computes the matching score from attention maps from different sampled timesteps in the diffusion steps from noisy image to clean image. This approach allows us to integrate both high-level and low-level features, as well as coarse and detailed aspects of image representations, which are crucial for effective image-text matching.
>
> We applied different noise levels during inference to get the score and ensemble the predictions results from each noise level to get the final prediction, which has been shown in our ablations to achieve the best performance.  This is because utilizing various input noise levels enables our model to enhance its robustness and adaptability across discriminative tasks. We have expanded the method section to clearly explain the role and motivation for using different noise levels. We have also updated Algorithm 1, where we explicitly include noise as an input parameter.
>
> Additionally, in Section E of the Appendix, we tested the setting that directly uses latent space features after the VAE for image-text retrieval without using noise as inputs. The performance was obviously lower than with noise input, but it demonstrates the potential of using Discffusion’s latent space features directly.
>
>
> **Inference Time**
>
> Our approach represents an initial foray into adapting pre-trained generative diffusion models for discriminative tasks. Our method is a general approach that can be applied to other faster diffusion models.  In Appendix Section C, we demonstrate the compatibility of our method with faster diffusion models, such as Latent Consistency Models.
>
> In general, it takes approximately 0.6 seconds per example with 100 sampling timesteps and 0.3 seconds with 50 sampling steps using Discffusion. For the CLIP baseline, it takes about 30 ms per example. In Appendix C: Adapting Discffusion to Latent Consistency Models, we present results on Latent Consistency Models, which take about 70 ms per example. Additionally, we have now tried our method on the Phased Consistency Model [1], which takes only 24 ms per example, making it even faster than the CLIP baseline and the performance is also better: 78.15% (Discffusion) vs. 76.10% (CLIP) accuracy on ComVG under the same setting. We will include the full results using PCM in the revision.
>
>
> [1] Phased Consistency Model
>
>
>
>
>
>
> **Classifier Guidance**
>
> We want to clarify that our original statement implies the classifier guidance involves training an additional classifier, and then adding a gradient term derived from the log probability of the class given an image using this trained classifier. This gradient is then incorporated into the original diffusion process during inference to guide the model towards generating images that more closely align with the specified class.
>
>
> Our Dynamic Attention Head Weighting is motivated by this approach. We use the score computed by each attention head to predict the class and compute a loss based on the distance between the predicted class and the ground truth. The gradient of this loss is then used to adjust the weight for each attention head, guiding the matching process.
>
> We have revised our paper to clarify these points and avoid any confusion.
>
>
> **Revised Conclusion**
>
> Our approach can better capture the fine-grained details, spatial relationships, and compositionality by adapting from pretrained Stable Diffusion, this is shown in our superior performance on Refcocog, ComVG, Winoground and VL-checklist dataset. Discffusion also has better explainability: during the diffusion process, the correspondence between the text and image is reflected in token-level scores on the attention map. Tokens receiving higher scores become predominant in the decision-making process for matching (as shown in Figure 10 of the Appendix). This enhances understanding and offers clear insight compared to traditional methods. We have revised the conclusion in the revision.

---

> ### Author Response · Authors · 2024-06-24
> **Author Response to Reviewer 2hXW (2/2)**
>
> **Comparison to Current SOTA CLIP Models**
>
> We have now added additional results from SigLiP on ComVG, RefCOCOG, and VQAv2 shown in the below table.
>
> | Model                                  | ComVG | RefCOCOg | VQAv2 |
> |----------------------------------------|-------|----------|-------|
> | CLIP                                   | 76.10 | 69.88    | 17.86 |
> | BLIP2*                                 | 80.81 | 76.43    | **20.99** |
> | OFA*                                   | 76.29 | 73.31    | 20.01 |
> | SigLiP                                   | 79.91 | 76.88    | 20.17 |
> | Discffusion (w/o pre-training)  | 78.11 | 73.07    | 20.43 |
> | Discffusion (w/ pre-training) | **80.89** | **76.91** | 20.83 |
>
> As noted in the main paper, state-of-the-art pre-trained vision-and-language models such as BLIP2 may include images from Visual Genome (VG), RefCOCOg, and VQAv2 during their pre-training. To ensure a fair comparison, we have now included and presented results from Discffusion with pre-training on these datasets. The resulting best performance on ComVG and RefCOCOg, along with competitive performance on VQAv2, highlights Discffusion is comparable against current SOTA CLIP models.
>
>
>
> **Generalization to Diffusion Transformer Architectures**
>
> Thank you for the suggestion. Discffusion is a general pipeline that can be adapted to different types of diffusion models. We implemented Discffusion on DiT XL/2 with a 512x512 image resolution (DiT-XL-2-512). We fine-tuned the model on the ComVG and RefCOCOg datasets, achieving the following results:
>
> **Compositional Visual Genome**
>
> | Method                 | Subjects | Objects | Predicate | Average |
> |------------------------|----------|---------|-----------|---------|
> | CLIP (Fine-tuning)     | 80.77    | 82.49   | 60.50     | 76.10   |
> | CLIP (Prompt Learning) | 78.88    | 79.51   | 60.41     | 74.24   |
> | Discffusion (DiT)     | 73.91| 74.48| 55.12| 69.22|
> | Discffusion (Stable Diffusion)     | **80.78**| **84.90**| **63.43**| **78.27**|
>
>
> **RefCOCOg**
>
> | Method                 | Top-1 Acc. | Top-5 Acc. |
> |------------------------|------------|------------|
> | CLIP (Fine-tuning)     | 69.88      | 84.57      |
> | CLIP (Prompt Learning) | 69.40      | 84.48      |
> | Discffusion (DiT)     | 62.11  | 78.96  |
> | Discffusion (Stable Diffusion)     |  **75.87**|  **91.96**|
>
>
> The performance with DiT is lower than with Stable Diffusion. This difference is likely due to the distinct handling of the noised latent input. Discffusion with Stable Diffusion processes the entire noised latent input and uses cross-attention maps between the features of the entire noised latent input and text features to compute the matching score. In contrast, DiT patchifies the noised latent input into a sequence of tokens by linearly embedding each patch, which may result in less accurate score computations. Additionally, the transformer architecture used in DiT involves a larger number of parameters to fine-tune in Discffusion (DiT), which may make the adaptation more challenging when the amount of available data is limited (5% data in our setting). Nevertheless, the results demonstrate the potential of Discffusion on DiT. With proper modifications, the performance can be further improved, validating the generalizability of Discffusion.
>
>
>
>
> **Consistency in Results Presentation**
>
> We have now standardized the formatting across all tables in the revision. The best results are consistently highlighted in bold to improve readability and ease of comparison.
>
>
>
>
> Overall, we thank you once again for your insightful comments and hope that our responses and revisions can address your concerns.

---

### Review · Reviewer_NNvx · 2024-06-10

**Summary Of Contributions:**

The paper introduces a novel method, Discriminative Stable Diffusion (Discffusion), which repurposes pre-trained text-to-image diffusion models for few-shot discriminative tasks, specifically image-text matching. The method leverages the cross-attention scores from pretrained diffusion models to evaluate the alignment between visual and textual information. By using additional learnable prompts, the model is fine-tuned to perform image-text matching with limited new data.

**Audience:**

Yes

**Broader Impact Concerns:**

Based on the information provided, it seems the authors have already addressed any potential broader impact concerns within their work.

**Claims And Evidence:**

Yes

**Requested Changes:**

1. While the authors introduce several techniques such as learnable prompts and log-sum-exponential pooling, it is challenging to evaluate how each component contributes to the overall performance. Additionally, the paper does not adequately address how these techniques generalize across different datasets. Moreover, the utility of the zero-shot setting in model selection is unclear.

2. The baseline methods used in this work are not clearly defined. For instance, the type of CLIP model utilized is unspecified. In comparison, OpenCLIP has been shown to provide better results in other studies [1]. It is unclear why OpenCLIP was not used as a baseline.

3. Table 3 shows that well-pretrained models like BLIP already achieve strong performance. It remains unclear how the proposed method can close the performance gap. For example, it would be insightful to understand how the proposed method would perform if it were also pretrained on datasets like RefCOCO.



[1] Li, Alexander C., et al. "Your diffusion model is secretly a zero-shot classifier." Proceedings of the IEEE/CVF International Conference on Computer Vision. 2023.

**Strengths And Weaknesses:**

Strength:

1. The proposed method demonstrates impressive performance across various benchmark datasets, highlighting its effectiveness.

2. The methodology is well-explained and straightforward, making it easy to understand and implement.

Weakness:

1. The ablation study lacks thoroughness, which leaves some aspects of the method's performance unexplored.

2.  The paper does not clearly define or compare its results against established baselines. Some of the baselines are not discussed.

---

> ### Author Response · Authors · 2024-06-24
> **Author Response to Reviewer NNvx (1/2)**
>
> Thank you for your constructive feedback and for recognizing the novel contributions, effectiveness, straightforwardness of our method. We appreciate the opportunity to clarify and extend the explanations of our methods and results.
>
>
>
>
> **Ablation Study**
>
> We acknowledge the need for a thorough ablation study. The Section 5.9 of our paper details extensive evaluations regarding different techniques used in Discffusion, including:
> - **Attention Maps from Different Sets of U-Net Layers**, demonstrating that using attention maps from all layers yields the best results.
>  - **Cosine Similarity vs. Maximum vs. LogSumExp Pooling** for score computation, where LogSumExp Pooling was found to be superior.
> - **Dynamic Attention Head Weighting** and its impact on performance. Our evaluations demonstrate that Dynamic Attention Head Weighting significantly enhances overall performance compared to using simple averaging across selected heads.
> - **Noise Level Ensembling**, showing the benefits of using noise level ensembling. It achieves improvements over individual noise levels of 0.4 and 0.8.
>
> Further, we direct you to Section A and F in the Appendix, which includes ablations on:
> - **Input Image Resolutions**, showing that Discffusion adeptly harnesses the benefits of larger input resolutions.
> - **Masking Out Objects during inference in RefCOCOg**, demonstrating how Discffusion captures key objects in images for accurate matching.
> - **Time Steps Sampling in Discffusion**, illustrating that leveraging all time steps’ information effectively integrates and processes the coarse-to-fine details throughout the diffusion steps, leading to enhanced performance.
> - **Loss Functions**, suggesting that employing the original margin-based triplet loss function outperforms alternative approaches. This superiority is primarily due to the original diffusion training loss’s inefficacy in adapting Stable Diffusion for discriminative tasks and the relative inadequacy of the cross-entropy loss in managing the nuances of image-text matching tasks.
>
>
> Regarding the **contribution of learnable prompts**:
>
> Without using learnable prompts and evaluating Discffusion in a zero-shot setting, as shown in Figure 5 (left), we achieve 75.4% accuracy on ComVG, which is lower than the 78.27% achieved by Discffusion with learnable prompts, as shown in Table 1. Additionally, we conducted additional experiments since the start of the rebuttal period where we directly fine-tuned the cross-attention matrices instead of using learnable prompts (w/o learnable prompts) . The results are as follows:
>
>
> **Compositional Visual Genome**
>
> | Method                 | Subjects | Objects | Predicate | Average |
> |------------------------|----------|---------|-----------|---------|
> | CLIP (Fine-tuning)     | 80.77    | 82.49   | 60.50     | 76.10   |
> | CLIP (Prompt Learning) | 78.88    | 79.51   | 60.41     | 74.24   |
> | Discffusion (w/o learnable prompts)     | 79.81| 82.47| 61.52| 76.44|
> | Discffusion     | **80.78**| **84.90**| **63.43**| **78.27**|
>
>
> **RefCOCOg**
>
> | Method                 | Top-1 Acc. | Top-5 Acc. |
> |------------------------|------------|------------|
> | CLIP (Fine-tuning)     | 69.88      | 84.57      |
> | CLIP (Prompt Learning) | 69.40      | 84.48      |
> | Discffusion (w/o learnable prompts)     | 73.56  | 90.08  |
> | Discffusion     | **75.87**|  **91.96**|
>
>
> The results indicate that directly fine-tuning the cross-attention matrices leads to lower performance, and it also brings longer training times and number of trainable parameters compared to using learnable prompts. This demonstrates the effectiveness of using learnable prompts techniques in Discffusion.
>
>
> **Generalization Across Datasets**
>
> We want to kindly refer you to our experiment section. Our experimental results across five diverse datasets (ComVG, RefCOCOg, VQAv2, Winoground, and VL-checklist) demonstrate the robust generalization capabilities of our proposed techniques.
>
>
> **Zero-shot Setting in Model Selection**
>
> As detailed in Section 5.9 of the Ablation Studies, we evaluated the effects of various settings under a zero-shot scenario. This includes exploring attention maps from all U-Net layers, using LogSumExp pooling, dynamic attention head weighting, and ensembling over noise levels. All of these techniques contributed to the final model configuration, where we use the optimum model validated in the zero-shot setting.

---

> ### Author Response · Authors · 2024-06-24
> **Author Response to Reviewer NNvx (2/2)**
>
> **Choice of CLIP Model**
>
> Regarding the choice of the CLIP model, as outlined in Appendix J (Hyper-parameter Settings), we utilized Stable Diffusion with the CLIP-ViT/H as the vision encoder. Therefore, for a fair comparison, we also use the CLIP-ViT/H-14 as the baseline which is the OpenCLIP version. We have made this more clear in the revision.
>
> **Pre-training on Datasets like RefCOCO**
>
> Our proposed method, Discffusion, is a generalizable approach that can be applied to various diffusion models. Leveraging stronger diffusion models with larger pre-trained datasets can enhance performance. In Table 5, we initially presented results using only 5% of the training data to demonstrate the model's few-shot learning capability. Given that OFA’s pre-training datasets include Visual Genome (VG), RefCOCOg, and VQAv2, and BLIP2’s pre-training datasets include Visual Genome (VG) and COCO, we have now expanded our experiments to include training on the entire datasets of ComVG, RefCOCOg, and VQAv2 for a fair comparison. The results are now as follows and show that our method outperforms other state-of-the-art methods on ComVG and RefCOCOg, and remains competitive on VQAv2. Besides, Discffusion with pre-training outperforms the one without pre-training, showing the potential of Discffusion using diffusion models pre-trained on larger datasets.
>
>
>
> | Model                                  | ComVG | RefCOCOg | VQAv2 |
> |----------------------------------------|-------|----------|-------|
> | CLIP                                   | 76.10 | 69.88    | 17.86 |
> | BLIP2*                                 | 80.81 | 76.43    | **20.99** |
> | OFA*                                   | 76.29 | 73.31    | 20.01 |
> | Discffusion (w/o pre-training)  | 78.11 | 73.07    | 20.43 |
> | Discffusion (w/ pre-training) | **80.89** | **76.91** | 20.83 |
>
> We have now included the pretraining results in the revision.

---

> > ### Comment · Reviewer_NNvx · 2024-07-09
> >
> > Thank you for your hard work in conducting additional experiments. All my questions have been thoroughly addressed.

---

> ### Author Response · Authors · 2024-07-10
> **Thank you!**
>
> Thank you again for your feedback and for acknowledging our efforts, and wish you a good day.
>
> Best,
>
> The authors

---

### Author Response · Authors · 2024-06-24
**General Response to All Reviewers**

Dear Reviewers,

We sincerely thank you for your thorough and insightful reviews of our paper! Your valuable feedback has greatly helped in improving the quality and clarity of our work. We have carefully considered each of your comments and made substantial revisions to address your concerns.

We are also pleased to inform you that we have uploaded a new version of the manuscript, with all changes and improvements highlighted in blue for easy identification. We have also provided a point-by-point response to each of your comments, addressing the specific issues raised and explaining how we have revised the paper accordingly.

Please feel free to contact us if you have any further questions or need additional clarification.

Sincerely,

Authors of Paper 2556

---

### Decision · Action_Editor_TTnu · 2024-08-12

**Recommendation:** Accept with minor revision

**Comment:**

The major concern from the reviewers is on the clarity of the paper presentation (see comments from Reviewers 2hXW and yyhJ). Additional experiments were added and the authors also included revised presentation of the algorithm lists for clarification. The revision addressed most of the concerns, although I would suggest in final revision the authors should consider comments from yyhJ and make the descriptions of the method precise. Also providing intuitions on each of the components of the method will be much helpful, especially if the explanations can be backed by the ablation studies.

**Audience:**

ML researchers working on diffusion models and generative classifiers.

**Claims And Evidence:**

The paper presents Discffusion, a technique that repurposes pre-trained text-to-image diffusion models for classification tasks. The idea is to fine-tune a pre-trained text-to-image diffusion model so that the cross-attention score between the image representation and the target text can be used for classification. Experiments tested this idea by experimenting on various few-shot classification datasets and comparing with baseline methods such as CLIP and diffusion classifier. Overall the proposed approach outperforms the baselines on many tasks. Ablation studies on the efficacy of the components of the method are also provided.

---

> ### Author Response · Authors · 2024-08-28
>
> Dear Action Editor,
>
> We have uploaded the deanonymized camera-ready version of our manuscript. In the final version, we have incorporated all the comments and suggestions, such as enhancing the method section with additional intuitions and explanations. We believe these changes have significantly improved the clarity and overall presentation of the paper.
>
> Thank you for your guidance and support throughout the review process!
>
> Best regards,
>
> The Authors